

# An optimized LSTM-based approach applied to early warning and forecasting of ponding in the urban drainage system

Wen Zhu[1], Tao Tao[1], Hexiang Yan[1], Jieru Yan[1], Jiaying Wang[1], Shuping Li[1], Kunlun Xin[1]

[1]College of Environmental Science and Engineering, Tongji University, Shanghai, 200000, China

*Correspondence to: Tao Tao (taotao@tongji.edu.cn)*

**Abstract.** An optimized LSTM-based approach applied to early warning and forecasting of ponding in the urban drainage system is proposed in this study. This approach can identify locations and process of ponding quickly with relatively high accuracy. The model is constructed with two tandem processes and a multi-task learning mechanism is introduced. The results are compared with those of widely used neural networks (LSTM, CNN) to validate its advantages. Then, the model is

revised with available monitoring data in the study area to achieve higher accuracy, and the influence of the number of the monitoring points selected on the performance of the corrected model is also discussed in this paper. Over 15000 designed rainfall events are used for model training, covering a diversity of extreme weather conditions.

## 1 Introduction

The intensity and frequency of urban floods are growing as a result of the increased frequency of extreme weather, rapid

urbanization, and climate change (Hossain Anni et al., 2020). It is becoming increasingly clear that urban floods have a significant impact on city management and endanger the safety of people's life and property. The ability to reliably characterize and forecast urban floods and generate high-precision flood risk maps from them has become critical in flood mitigation and decision making.

The most common approach to simulating urban floods is to develop a hydrodynamic model, which uses the collected

topographic map, information on pipe network, historical rainfall data, monitoring data, and other information in the study area (Jamali et al., 2018; Aryal et al., 2020; Balstrøm and Crawford, 2018; Tian et al., 2019). However, a realistic hydrodynamic model for continuous simulation necessitates a huge quantity of data, such as comprehensive topography, infiltration conditions, and sewage system data, including exact locations, depths, and diameters of sewage pipes, all of which are difficult to obtain in metropolitan areas (Rahman et al., 2002; Kuczera et al., 2006). Furthermore, correlative

calculation in storm-inundation simulation is sophisticated, often computationally intensive, and takes a long time to execute. The most detailed representation of the storm-inundation simulation is the 1D-2D model (Djordjević et al., 1999; Djordjević et al., 2005), which summarizes the dynamic interaction between the flow that enters the underground drainage network and the overloaded flow that spreads to the surface flow network during high-intensity rainfall. Some of the available models that



realize storm-inundation simulation include XPSWMM, TUFLOW, and MIKE FLOOD (Leandro and Martins, 2016; Teng
et al., 2017; Zhang and Pan, 2014).

All of the abovementioned problems have hampered the continuous development of the hydrodynamic models in urban flood
forecasting. As a result, machine learning, particularly deep learning, has emerged as another viable forecasting tool. Deep
learning is a form of training that involves creating a unique dataset and then analyzing and forecasting urban flooding using
algorithms (Mudashiru et al., 2021; Sit et al., 2020; Shen, 2018). It can compensate for the impact of actual data scarcity by
training on a large designed data set and does not require any assumptions on the physical processes as required by the
traditional hydrodynamic models.

However, some factors have hampered the application of deep learning in urban flood forecasting. Firstly, the data set for
training is insufficient to reflect the superiority of the approach, for example, Cai and Yu, (2022) used only 25 historical
floods for forecasting calculations, and Abou Rjeily et al., (2017) used 10 rainfall events for training and verification, which
are insufficient to reflect the characteristics of rainfall distribution. Secondly, monitoring equipment is expensive and thus
not frequently available, thus researchers have to rely on simulations produced from hydrodynamic models without regard
for their accuracy. For example, Chiang et al., (2010) used synthetic data produced from SWMM as the target values to train
the recurrent neural network (RNN) and compared the predictions with simulation results to evaluate the accuracy of the
model in estimating water levels at ungauged locations. Thirdly, to improve the performance of the model, some studies, e.g.,
the runoff forecasting tasks with multiple time steps, have focused on building more complex deep learning architectures,
such as the automatic encoder (Bai et al., 2019), encoder-decoder (Kao et al., 2020c), and customized layers based on Long
Short-Term Memory (LSTM) (Sit et al., 2020; Kratzert et al., 2019; Kratzert et al., 2019). For example, an encoder-decoder
LSTM was proposed for runoff forecasting for up to 6 hours and 24 hours ahead (Xiang et al., 2020b; Kao et al., 2020c).
Nevertheless, the urban flooding forecasting tasks with multiple time steps are mainly based on the precipitation forecast
hours in advance, which is not discussed in this paper. With the real-time rain data in a short duration, it is not available to
get enough data like the continuing runoff data to support the hours ahead prediction.

The objective of this study is to propose an optimized LSTM-based approach applied to early warning and forecasting of
ponding in the urban drainage system. This approach can identify locations and process of ponding quickly with relatively
high accuracy. The model was constructed with two tandem processes and a multi-task learning mechanism was introduced.
The results of the optimized LSTM-based model were compared with those of widely used neural networks (LSTM, CNN)
to validate its advantages. The model was revised with available monitoring data in the study area to achieve higher accuracy,
and the influence of the number of the monitoring points selected on the performance of the model in the model updating
procedure was discussed in this paper. Over 15000 designed rainfall events were used for model training, covering a
diversity of extreme weather conditions.

The rest of the paper is organized as follows: Section 2 introduces the methodology including the proposed LSTM-based
modeling framework; the experimental setup and application are also explained in this part; Section 3 and 4 present and
discuss the results respectively; and finally, brief conclusions are drawn in Section 5.



## 2 Methodology

### 2.1 LSTM-based model

Based on how the existing hydrodynamic models work, where surface runoff and pipeline confluence can be calculated with different hydrodynamic or hydrologic mechanisms, the LSTM-based model proposed in this study is constructed with two stages, runoff process and flow confluence process, to reduce the computational burden of this data-driven model. Fig. 1 presents the model architecture from input to output for both runoff and flow confluence processes.

The two processes are in tandem: the inputs of the flow confluence process are inherited and concatenated from the outputs

of all nodes in the runoff process. However, during the training process, the two processes are trained separately and without mutual interference when the inputs and outputs of both processes are the simulated results.

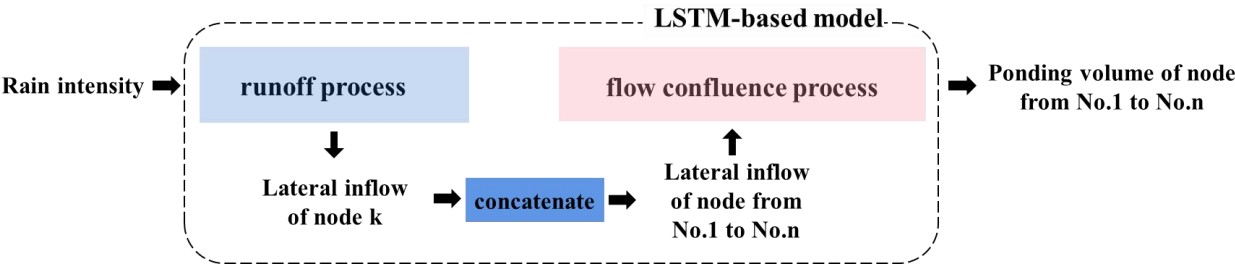

**Figure 1: The architecture of the LSTM-based model.**

### 2.1.1 Runoff process

With a general understanding of the hydrological mechanism of hydrodynamic models, the task of the runoff process is to simulate the external inflows of the pipe network which involves surface runoff and infiltration, and the most important influencing factor is rainfall. The mass rainfall curve reflects the characteristics of a specific rainfall process, thus it can be directly used as the input of a neural network. Meanwhile, lateral inflows at each node reflect the hydraulic state in the runoff process and are used as the outputs.

Fig. 2 illustrates the runoff process's training, verification, and test processes in detail. As illustrated in Fig.2, the neural network is fed with a training set that is constituted with two time-series data, i.e., rainfall intensity and lateral inflows that enters each node. At each epoch, four indicators (i.e., MAE, MSE, CC, NSE) are used to evaluate the consistency between the predicted lateral inflows and the simulation from the hydrodynamic model to verify whether the network meets the requirement for convergence. If the model converges, the network is evaluated on the test set, otherwise, go to the next

epoch for training. In the figure, MAE is short for Mean Absolute Error, MSE represents Mean Squared Error, CC is Correlation coefficient and NSE is Nash-Sutcliffe efficiency coefficient. S represents the advance time step for prediction which is fixed at 5min in this paper because of the interval for the measured data obtained.



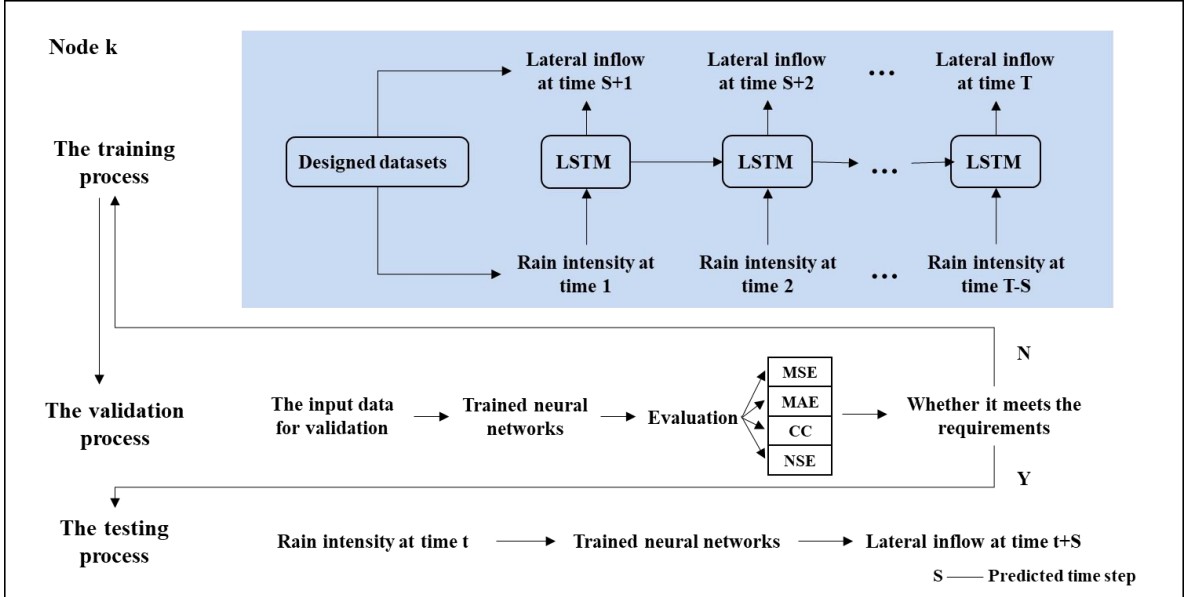

**Figure 2: The training, verification, and test process in the runoff process.**

## 2.1.2 Flow confluence process

With the understanding of the simulation process of one-dimensional confluence in the pipe network of a hydrodynamic model(e.g., the SWMM model), if the whole urban drainage system is regarded as a black box, mainly the lateral inflows at each node and outflows from the outlets enter and leave the system, respectively. Therefore, the inputs of the flow confluence process are the lateral inflows at each node and the outflows. If the outflows are set free, the hydraulic state behind the outlets has little influence on the interior of the case area, and the outflows could be removed from the inputs.

To improve the model performance, the network architecture in the flow confluence process is optimized with an auxiliary task. Fig. 3 illustrates the details of network architecture in the flow confluence process. First of all, a Gaussian layer is added after the input layer in the flow confluence process to avoid the interference caused by the training error in the runoff process. Then, a classification task is introduced as an auxiliary task to the ponding volume forecasting. The classification module is added after the outputs of each time step in the LSTM module to judge whether ponding occurs at the time. Only if ponding has occurred at this time, can the output of the LSTM module at the time step enter the 'OUT_MODULE' (as shown in Fig. 3) to continue learning; otherwise, discard the output of the LSTM module at this time step. In this way, the interference of the time points without ponding on the ponding volume forecasting is eliminated to the greatest extent. The higher accuracy of the classification, the more accurate the prediction of ponding volume will be. Moreover, because of the hard sharing mechanism of parameters in multi-task learning, a part of parameter layers in the LSTM module are shared by the classification module and the out module, which effectively alleviates the over-fitting of the model.





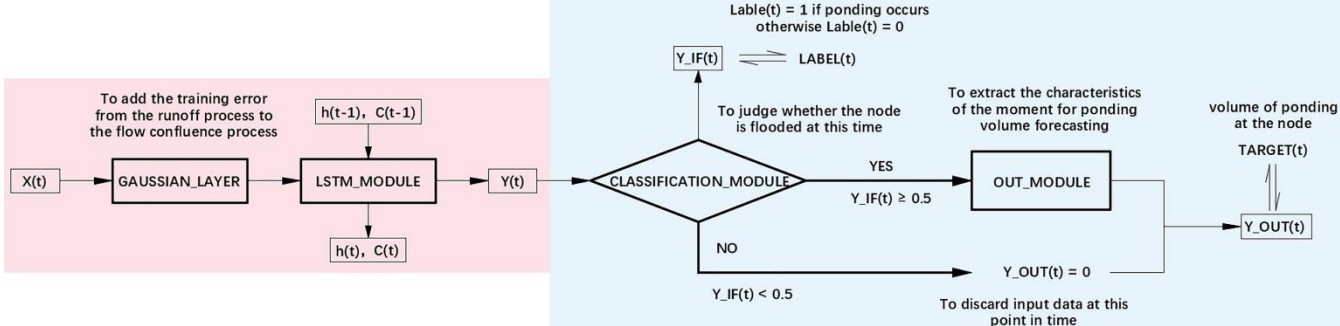

**Figure 3: The LSTM module with an auxiliary task in the flow confluence process.**

## 2.2 Error transmission

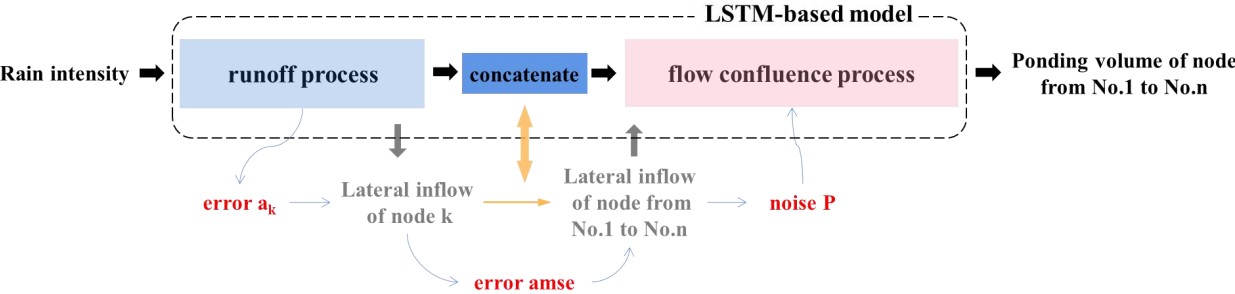

110

**Figure 4: Flow chart of the error transmission process.**

It is worth noting that simulation results from the hydrodynamic model are not used during testing: the inputs of the neural network in the flow confluence process are the outputs of the trained neural network in the runoff process. Thus the training error from the neural network in the runoff process will inevitably interfere with the neural network in the flow confluence

115 process, i.e., the error propagates.

Fig. 4 illustrates the details of the error transmission process from the runoff process to the flow confluence process during training. To avoid the interference caused by the training error in the runoff process and effectively alleviate the over-fitting of the neural network, noise P ($P \sim N(0, p^2)$) is added to the lateral inflows before the data is fed to the neural network in the flow confluence process. The magnitude of noise can be roughly determined as follows.

120 1 The mean square error (MSE) is used to characterize the training error in the runoff process. The error at node K is computed as follows:

$$a_k = \frac{\sum\limits_{i=1}^{T}\sum\limits_{j=1}^{S}\left(\widehat{X}_{ij} - X_{ij}\right)^2}{T \cdot S} \qquad (1)$$



where T represents the duration of event j in min, S represents the number of events in the training data, $\widehat{X}_{ij}$ represents the simulated lateral inflow at node K of the i-th time step in the j-th rainfall events in L/s, $X_{ij}$ represents the output of the runoff process at node K of the i-th time step in the j-th sample event in L/s.

2 Then compute the average mean square error of all nodes as follows:

$$\text{amse} = \frac{\sum_{k=1}^{N} a_k^2}{N} \tag{2}$$

where N represents the number of nodes.

3 Then convert amse into ε with the mean value of the predicted lateral inflows at all nodes in the training set. Eq. (3) is the definition of noise percentage(ε) and the relationship of amse and the noise percentage:

$$\varepsilon = \sqrt{\frac{P_N}{P_S}} = \sqrt{\frac{\sum_{k=1}^{N} \sum_{i=1}^{T} \sum_{j=1}^{S} \left(\widehat{x}_{kij} - x_{kij}\right)^2}{\sum_{k=1}^{N} \sum_{i=1}^{T} \sum_{j=1}^{S} \left(x_{kij}\right)^2}} \leq \sqrt{\frac{\sum_{k=1}^{N} \sum_{i=1}^{T} \sum_{j=1}^{S} \left(\widehat{x}_{kij} - x_{kij}\right)^2}{\frac{1}{N \cdot T \cdot S} \left(\sum_{k=1}^{N} \sum_{i=1}^{T} \sum_{j=1}^{S} x_{kij}\right)^2}} = \frac{\sqrt{\text{amse}}}{\frac{1}{N \cdot T \cdot S} \sum_{k=1}^{N} \sum_{i=1}^{T} \sum_{j=1}^{S} x_{kij}} \tag{3}$$

where $P_S$ represents signal power, and $P_N$ represents noise power.

4 Add noise P to the inputs (X) in the flow confluence process during the training process. Firstly, generate a set of random numbers G with the length of X by Pseudorandom Number Generator which obeys a normal distribution (G~N(0,1)), i.e., $P = p \cdot G$.

$$p = \varepsilon \cdot \sqrt{\frac{1}{T} \sum_{i=1}^{T} (X_i)^2} \tag{4}$$

## 2.3 Model correction system

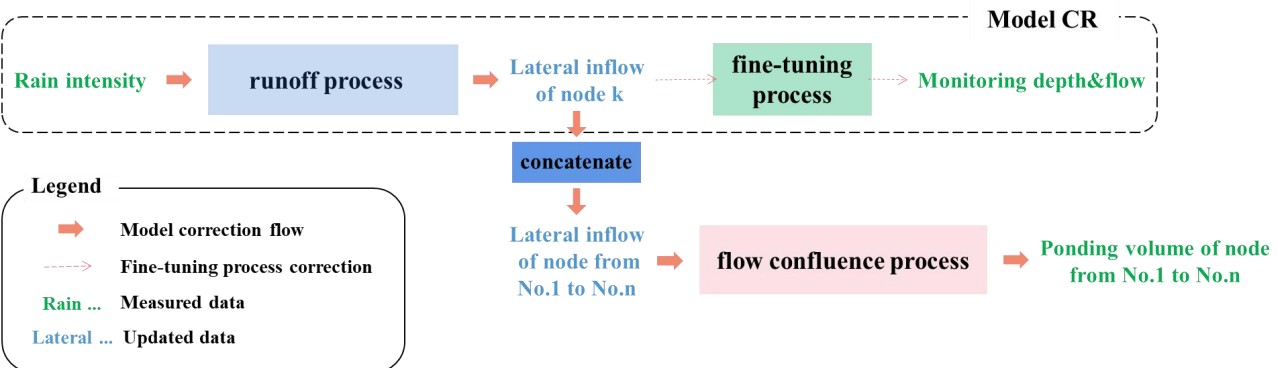

**Figure 5: Model correction system.**

The LSTM-based model is constructed based on a relatively accurate hydrodynamic model. However, the difference between the simulation from the hydrodynamic model at the monitoring points and the actual measured monitoring data persists during the pipe network operation, leading to a discrepancy between the predicted results from the proposed LSTM-based model and the actual situation. This problem can be alleviated by correcting the model with the measured level and flow data at the monitoring points. The way to revise the model with the measured monitoring data is one of the focuses of



this study. The LSTM-based model is corrected with two steps. Fig. 5 indicates the model correction process using the

measured rain data, monitoring data obtained from the monitoring points, and ponding data at any node.

Firstly, the runoff process is corrected with the measured rain, level, and flow data according to the relationship between the

rain intensity and the monitoring data in a corresponding rainfall event referring to the transfer learning (PAN, S J, et al.,

2010). Transfer learning is to transfer the knowledge of one domain (source domain) to another domain (target domain) so

that the target domain can achieve better learning effects.

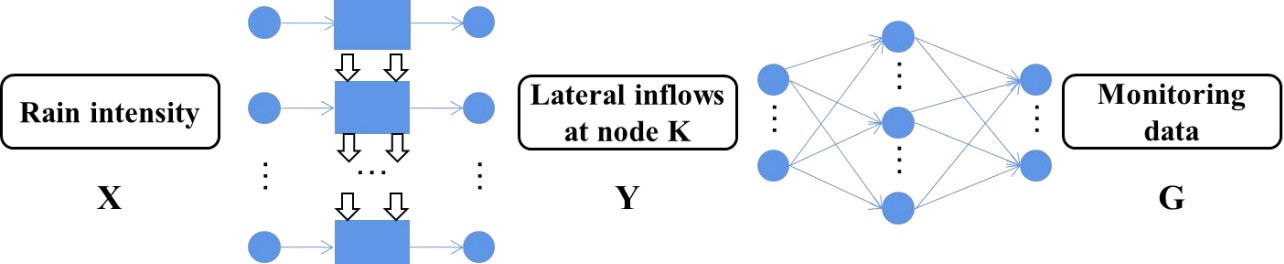

**Figure 6: The architecture of Model CR.**

Fig. 6 showed the schematic of the model CR. The network structure in the runoff process from rain data (X) to lateral

inflows (Y) will be migrated to the input-output connection between rain data and monitoring data (G). Multiple fully

connected layers are further added after the output layer of lateral inflows of one node to establish the model CR, which

contains the original runoff process and the additional fine-tuning process.

In the setup process of model CR, the original network structure in the runoff process keeps the characteristics obtained

from rain data to lateral inflows unchanged. Meanwhile, the variation of the lateral inflows at different nodes will cause

different hydraulic responses at the monitoring points to make sure that the neural network in the fine-tuning process will be

trained well. During the training process, the original parameters in the runoff process are fixed, only the parameter layers in

the fine-tuning process will be trained with the constructed rainfall data and simulated water depth and flow data at the

monitoring points.

Then, the model CR is updated using the measured rain data and monitoring data. Then, the outputs of the middle layer in

the updated model CR, i.e., the updated lateral inflows at each node, are obtained with the measured rain data as the input.

The flow confluence process is corrected with the updated lateral inflows of all nodes concatenated and the measured

ponding volume.

**2.4 Case study**

**2.4.1 Study area**

In the LSTM-based model, nodes are trained one by one. Both in the runoff process and flow confluence process, the

training process of different nodes will not interfere with each other. Due to these structural characteristics, the performance

of the model will not be limited by the size of the case area. Although a large-scale case area with more nodes will



significantly increase the time spent training the model and require extra processing power, it will not degrade the accuracy of the model on ponding forecasting.

To verify the feasibility of the modeling method above, a small-scale case area, JD, a residential district in S city, is selected as the study area. It is located in the northwest of S city. The elevation map of the study area is shown in Fig. 7(a). There are 32 residential buildings in the district, with a total area of 6.128hm$^2$. The study area is separated from the municipal roads by walls with three entrances that are located on the north, east and west sides of the community. Rain pipes laid in the study area are circular pipes, with diameters of 200mm, 300mm, 400mm, 500mm, and 600mm, most of which are 300mm in diameter. The total length of this pipe network is 5.5 km. The network contains 336 nodes and 340 pipes, which are

connected to the municipal pipe networks through 4 outlets as marked in green in Fig. 7(b). There are in total 15 level gauges and 3 flowmeters laid in the current pipe network. The layout of monitoring points is shown in Fig. 7(b).

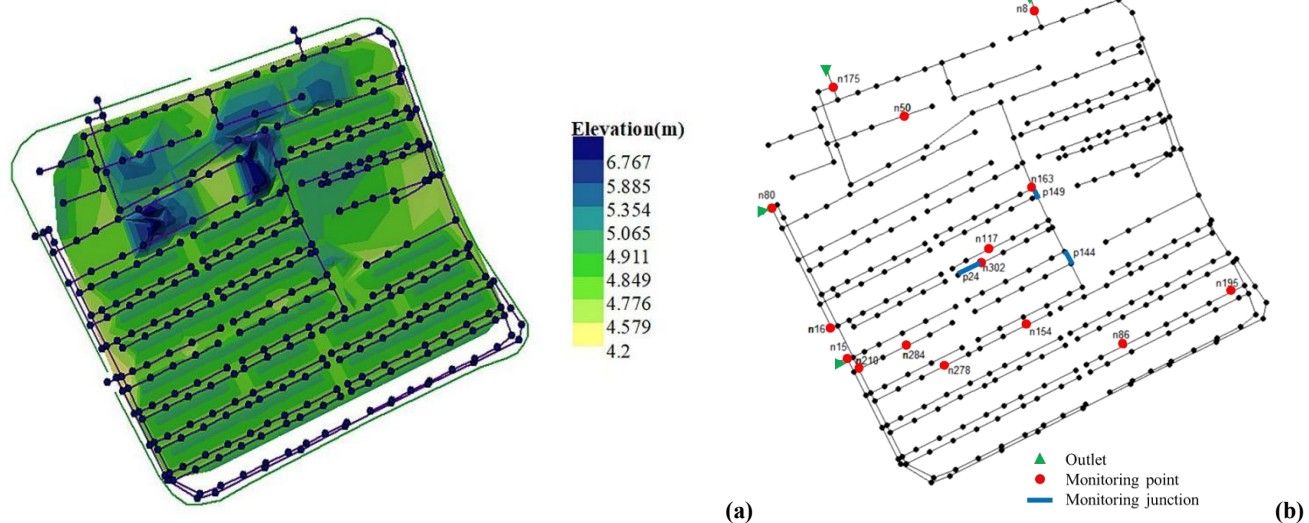

(a) The elevation map and stormwater system in the case area. (b) Locations of the monitoring points in the case area.

**Figure 7: Study area - JD residential district.**

**2.4.2 Rainfall data**

The rainstorm intensity for S city was computed by Eq. (5), and the rainstorm intensity before or after the peak was shown in Eq. (6). They were determined referring to the local rainstorm intensity formula, design storm pattern, and historical rain data.

Then single-peak rainfall scenarios were constructed unevenly by using different rainfall reappearing period (P) ranging

from 0.5a to 100a, peak coefficient (r) ranging from 0.1 to 0.9, and duration (T) ranging from 60 to 360 min.

$$q = \frac{167A(1+C\lg P)}{(t+b)^n} = \frac{1600(1+0.846\lg P)}{(t+7.0)^{0.656}} \qquad (5)$$





where q is the rainstorm intensity in L·s-1·hm-2, P is the rainfall reappearing period in a, t is the duration of rainfall in min, A, C, b, and n are parameters in the rainstorm intensity formula.

$$\begin{cases} I(t_b) = \dfrac{A(1+C\lg P)\left[\dfrac{(1-n)}{r}t_b + b\right]}{\left(\dfrac{t_b}{r}+b\right)^{n+1}} \\ \\ I(t_a) = \dfrac{A(1+C\lg P)\left[\dfrac{(1-n)}{1-r}t_a + b\right]}{\left(\dfrac{t_a}{1-r}+b\right)^{n+1}} \end{cases}$$

(6)

where $t_b$ and $t_a$ is the time before and after the peak in min respectively, and r is the rainfall peak coefficient.

In addition, according to the historical bimodal rainfall data in S city, the rainfall peaks corresponding to the bimodal design storm pattern in the duration of rainfall from 60 to 360 min could be computed by Pilgrim & Cordery. Then, double-peak rainfall scenarios were constructed where the rainfall reappearing period was from 0.5a to 100a.

**Table 1: The bimodal design storm pattern.**

**(a) Duration is 60 min.**

| T(5min) | P/Pmax(%) |
|:---:|:---:|
| 1 | 3.19 |
| 2 | 16.71 |
| 3 | 8.74 |
| 4 | 1.52 |
| 5 | 2.27 |
| 6 | 4.02 |
| 7 | 5.89 |
| 8 | 22.68 |
| 9 | 10.51 |
| 10 | 12.76 |
| 11 | 7.05 |
| 12 | 4.68 |

**(b) Duration is 120 min.**

| T(5min) | P/Pmax(%) |
|:---:|:---:|
| 1 | 0.74 |



| T(5min) | P/Pmax(%) |
|---|---|
| 2 | 1.76 |
| 3 | 6.42 |
| 4 | 3.75 |
| 5 | 2.05 |
| 6 | 1.52 |
| 7 | 2.73 |
| 8 | 4.43 |
| 9 | 9.23 |
| 10 | 11.17 |
| 11 | 17.57 |
| 12 | 13.81 |
| 13 | 7.76 |
| 14 | 5.3 |
| 15 | 2.38 |
| 16 | 1.13 |
| 17 | 3.17 |
| 18 | 1.32 |
| 19 | 0.98 |
| 20 | 0.84 |
| 21 | 0.54 |
| 22 | 0.64 |
| 23 | 0.44 |
| 24 | 0.33 |

Table 1 showed the bimodal design storm patterns respectively in 60 min and 120 min. P/Pmax in the table represented the distributions of the rainfall intensity over time. The unit period was 5 minutes.

To expand the size of the designed datasets and ensure the datasets contain a majority of extreme rainfall events, both single and double-peak rainfall data produced were added with Gaussian white noise. The noise percentages went from 0 to 100% at an interval of 10% to blur the characteristics of the design storm pattern and bring the rainfall closer to reality. The synthetic dataset contained a total of 16960 rainfall events. And the ratios of the training set, validation set, and test set were 80%, 10%, and 10%, respectively.



### 2.4.3 Simulated and measured data

A hydrodynamic model was established for the case pipe network, and the simulation results (i.e., the lateral inflows and the volume of ponding at each node, as well as the level and flow data at the monitoring points) were obtained by using the constructed rainfall events as described in Sect. 2.4.2. Both the constructed rainfall events and the simulated were used in the training, validation, and testing processes. In the simulation process, we considered a uniform distribution of rainfall in space. A simplified representation of the sewer system and a constant, uniform infiltration rate in the green area were considered for

runoff computation (Roland L¨owe et al.,2021). Meanwhile, the two-dimensional surface overflow process is not considered in this paper.

  Besides, the actual rain data and the measured water depth and flow data at the monitoring points placed in the study area of the past 5 rainfall events were obtained to verify the performance of the corrected model. In the model correction process, the uncertainty of the measurement (i.e. rainfall data, water depth, and flow data at monitoring points) was not considered,

for example, errors in data transmission of monitoring equipment.

  Table 2 showed 5 measured rainfall events used in the model correction system. Three of them were used to correct the model CR and the flow confluence process, and the others were used to evaluate the reliability of the approach to correcting the model.

**Table 2: 5 rainfall data used to correct the LSTM-based model.**

| Dataset | Rainfall event | Rainfall (mm) | Max. rain intensity (mm/min) | Duration (min) |
|---------|----------------|---------------|------------------------------|----------------|
| Training | No.1 | 494.50 | 8.13 | 180 |
|          | No.2 | 146.63 | 2.61 | 90 |
|          | No.3 | 254.51 | 3.61 | 240 |
| Testing  | No.4 | 442.61 | 7.26 | 150 |
|          | No.5 | 254.41 | 4.97 | 120 |

### 225 2.4.4 Model Construction

  Most of the hyper-parameters in this paper were determined by Hyperopt (BERGSTRA, J, et al., 2013). Hyperopt is a Python library for hyper-parameter optimization that adjusts parameters by Bayesian optimization, allowing optimal parameters to be obtained for a given model. The search space of hyper-parameters was artificially set by trial and error first. Then, receive a random combination of hyper-parameters from the search space to do the training and return the training loss.

Use the built-in search algorithms like the tree of Parzen estimators (TPE) to determine the next hyper-parameters combination. Thus, these iterations gradually build an optimal hyper-parameters combination to get the lowest training loss.



Hyper-parameters in the learning process of the model setup and model correction obtained by Hyperopt was shown in Table 3.

**Table 3: Hyper-parameters configuration in model setup and correction processes.**

|  | Hyper-parameters | Runoff process | Fine-tuning process | Model CR | Flow confluence process |
|---|---|---|---|---|---|
| | Normalization | Z-score | Z-score | Z-score | Min-Max |
| | Batch size | 150 | 150 | 150 | 150 |
| | Epoch | 300 | 300 | 300 | 300 |
| Model setup | Learning rate | 1e-2 | 5e-3 | | 1e-2 |
| | Optimizer | Adam | SGD | | SGD |
| | LSTM hidden layer neurons | 16 | - | | 256 |
| | Linear hidden layer neurons | 16 | 3072 | | 128 |
| | LSTM layers | 2 | - | | 4 |
| | Linear layers | 1 | 2 | | 2 |
| Model correction | Learning rate | | | 1e-4 | 5e-5 |
| | Optimizer | | | Adam | SGD |

### 2.4.5 Performance evaluation

MAE, MSE, NSE, and CC are always used to assess the performance of data-driven models. In this paper, MAE and MSE were used to quantify the errors between the predicted results by the neural network and simulation from the hydrodynamic model at each node. Moreover, NSE and CC here were used to evaluate the level of agreement at all nodes. These criteria were described in Eq. (7)-(10).

$$MAE = \frac{1}{DT} \sum_{s=1}^{D} \sum_{t=1}^{T} \left| Y_{st} - \hat{Y}_{st} \right| \tag{7}$$

$$MSE = \frac{1}{DT} \sum_{s=1}^{D} \sum_{t=1}^{T} \left( Y_{st} - \hat{Y}_{st} \right)^2 \tag{8}$$



$$NSE = 1 - \frac{\sum\limits_{t=1}^{T}\left(\frac{1}{D}\sum\limits_{s=1}^{D}Y_{st} - \frac{1}{D}\sum\limits_{s=1}^{D}\hat{Y}_{st}\right)^2}{\sum\limits_{t=1}^{T}\left(\frac{1}{D}\sum\limits_{s=1}^{D}\hat{Y}_{st} - \frac{1}{DT}\sum\limits_{t=1}^{T}\sum\limits_{s=1}^{D}\hat{Y}_{st}\right)^2} \tag{9}$$

$$CC = \frac{\sqrt{\sum\limits_{t=1}^{T}\left(\frac{1}{D}\sum\limits_{s=1}^{D}Y_{st} - \frac{1}{DT}\sum\limits_{t=1}^{T}\sum\limits_{s=1}^{D}Y_{st}\right)\left(\frac{1}{D}\sum\limits_{s=1}^{D}\hat{Y}_{st} - \frac{1}{DT}\sum\limits_{t=1}^{T}\sum\limits_{s=1}^{D}\hat{Y}_{st}\right)}}{\sqrt{\sum\limits_{t=1}^{T}\left(\frac{1}{D}\sum\limits_{s=1}^{D}Y_{st} - \frac{1}{DT}\sum\limits_{t=1}^{T}\sum\limits_{s=1}^{D}Y_{st}\right)^2}\sqrt{\sum\limits_{t=1}^{T}\left(\frac{1}{D}\sum\limits_{s=1}^{D}\hat{Y}_{st} - \frac{1}{DT}\sum\limits_{t=1}^{T}\sum\limits_{s=1}^{D}\hat{Y}_{st}\right)^2}} \tag{10}$$

where D is the number of events in the test set, T is the total time steps of the rainfall, $Y_{st}$ is the value predicted by the neural network at the t-th time step in the s-th event, and $\hat{Y}_{st}$ is the result from the hydrodynamic model.

Some other criteria shown in Table 4 were introduced to evaluate the judgment of whether a node was flooded as predicted by the model. Accuracy (ACC), Precision (PPV), and False omission rate (FOR) were used to verify the accuracy of the time of occurrence and duration when ponding occurred in a rainfall event. $S - PPV$ and $S - FOR$ were used to evaluate the accuracy of the event where ponding occurred.

**Table 4: Score values used for measuring the level of agreement between the judgment of whether ponding occurred by the neural network and simulation from the hydrodynamic model at one node.**

| Score | Purpose | Equation | Range | Best Value |
|-------|---------|----------|-------|-----------|
| ACC | Mean accuracy for time points classified correctly | $ACC = \frac{1}{D}\sum\limits_{s=1}^{D}\frac{TP_s + TN_s}{TP_s + TN_s + FP_s + FN_s}$ | $0-1$ | 1 |
| PPV | Mean precision for well-judged time point | $PPV = \frac{1}{D}\sum\limits_{s=1}^{D}\frac{TP_s}{TP_s + FP_s}$ | $0-1$ | 1 |
| FOR | Mean proportion of omission on the timeline | $FOR = \frac{1}{D}\sum\limits_{s=1}^{D}\frac{FN_s}{TN_s + FN_s}$ | $0-1$ | 0 |
| $S - PPV$ | Precision for well-judged samples | $S - PPV = \frac{TP}{TP + FP}$ | $0-1$ | 1 |
| $S - FOR$ | Percentage of samples with false negatives | $S - FOR = \frac{FN}{TN + FN}$ | $0-1$ | 0 |

where TP and TN denote the number of events where ponding occurs and does not occur that are correctly classified, FP and FN are the number of events where ponding occurs and events where ponding does not occur incorrectly classified, respectively. Besides, TPs and TNs are the number of time steps in the s-th event when ponding occurs and does not occur correctly classified, FPs and FNs are the number of time steps that are incorrectly classified.





# 3 Results

## 3.1 Model setup

The LSTM-based model was trained by designed rainfall data and simulation from the hydrodynamic model following the procedure described in Sect. 2.4. According to the steps in Sect. 2.2, the noise($\varepsilon$) transmitted from the runoff process to the
flow confluence process was calculated as 1.9412% in the case pipe network. For the convenience of calculations, the noise was set as 2%. The evaluation results of the performance of the trained model were shown as follows.

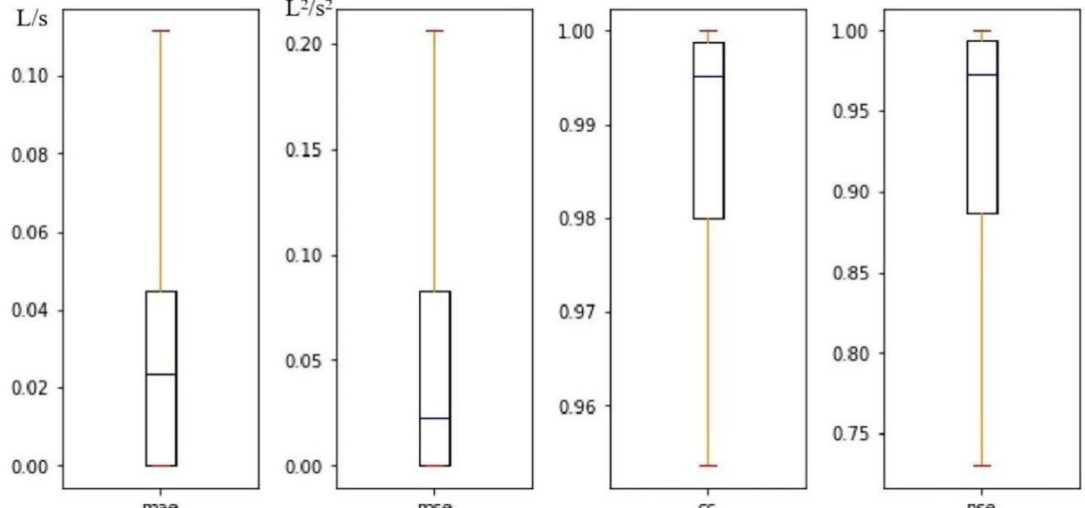

**Figure 8: Box plots of score values for comprehensive evaluation of all nodes in the case area in the model development procedure.**

By a comparison between the predicted ponding volume by the LSTM-based model and simulation from the hydrodynamic
model, Fig. 8 described the overall performance of the model with 4 box plots of mean scores on the test set of all nodes in the study area with the outliers removed. The median value of MAE and MSE were far less than 0.1, indicating that the training had converged at all nodes. The median value of CC was close to 1 while the minimum value was greater than 0.95, and the median value of NSE was greater than 0.95 while the minimum value was around 0.75, indicating that although the performance of the model at each node was slightly different, the overall prediction results of the model were reliable.
In this article, only the evaluation results of some representative nodes were listed due to the limited space. To illustrate the universality of the results in the pipe network, six nodes were randomly selected, and their locations were marked in Fig. 9.





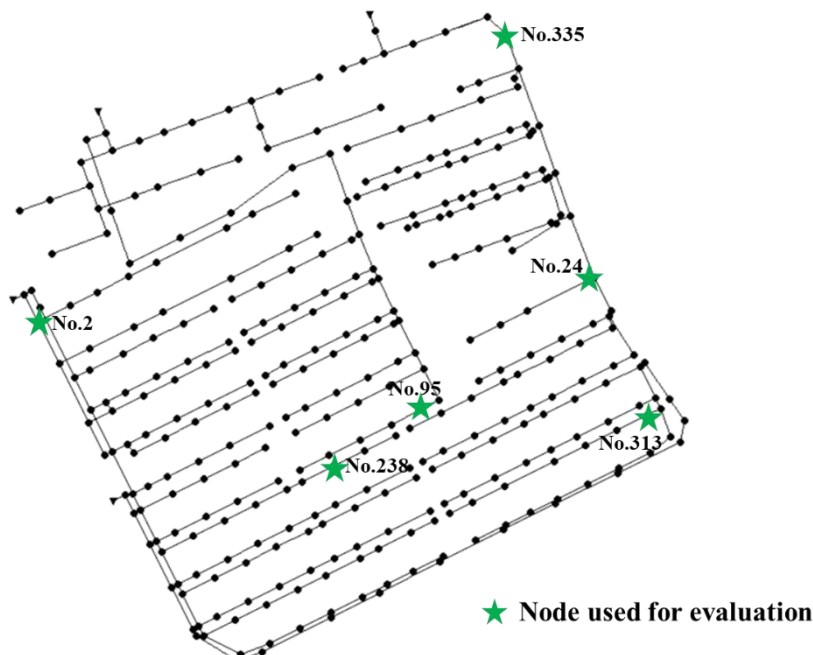

**Figure 9: The locations of 6 nodes for evaluation.**

On the premise that the model performed differently from node to node, Table 5 listed the score values of the six selected

nodes on the judgment of ponding, and Table 6 presented the scores on the volume forecasting.

**Table 5: Score values of the 6 selected nodes for the accuracy of the judgment of whether ponding occurred at some time step and in a rainfall event predicted by the model.**

| Node No. | ACC | PPV | FOR | S − PPV | S − FOR |
|---|---|---|---|---|---|
| 2 | 99.90% | 95.11% | 0.04% | 98.00% | 0.00% |
| 24 | 99.56% | 95.21% | 0.27% | 99.33% | 0.00% |
| 95 | 98.66% | 93.47% | 0.98% | 100.00% | 0.00% |
| 238 | 99.81% | 88.33% | 0.06% | 94.00% | 0.00% |
| 313 | 99.67% | 95.75% | 0.19% | 99.33% | 0.00% |
| 335 | 99.56% | 95.29% | 0.23% | 100.00% | 0.00% |

The second to fourth columns of Table 5 reflected the sensitivity of the model in judging the starting and ending time of

ponding. The accuracy of judging whether ponding occurred at some time step (ACC) of each node was above 98.5%.

Although the PPV of each node was slightly lower than ACC, it was higher than 88%, which indicated that about 12% of the

time points were judged incorrectly. FOR of each node was generally lower than 1%, and that of Node 95 was slightly

higher, which indicated that the model ignored ponding at some time points, but its proportion was far less than the false

discovery rate. The last two columns of Table 5 respectively showed that the model performed well at each node in judging





whether ponding occurred in the rainfall event. Node 238 falsely reported whether ponding occurred in 6% of the testing

events, resulting in the lowest $S - PPV$. $S - FOR$ in Table 5 indicated that the model did not miss any incidents where

ponding occurred during the rainfall events.

Node 95 had the highest MAE score 0.0770L/s, while the maximum MSE score was 0.3788L$^2$/s$^2$ at Node 2. CC scores

varied slightly between nodes which were close to 1. Node 238 had the lowest NSE score which remained higher than 0.8

though. Table 6 indicated that the predicted volume of ponding at each node was reliable.

**Table 6: Score values of the 6 selected nodes for the model performance on the ponding volume forecasting.**

| Node No. | MAE(L/s) | MSE(L$^2$/s$^2$) | CC | NSE |
|---|---|---|---|---|
| 2 | 0.017 | 0.3788 | 0.9997 | 0.9811 |
| 24 | 0.0414 | 0.1876 | 0.9941 | 0.9754 |
| 95 | 0.077 | 0.274 | 0.986 | 0.9599 |
| 238 | 0.0073 | 0.026 | 0.9999 | 0.8195 |
| 313 | 0.0183 | 0.0505 | 0.9974 | 0.9825 |
| 335 | 0.0349 | 0.0826 | 0.9968 | 0.9882 |

Furthermore, in the above analysis, mean score values on the test set were used for evaluation, and the individual differences

among the test set were ignored. Fig. 10 showed the predicted volume of ponding at the selected nodes as compared to

simulation results in several randomly selected testing rainfall events. As shown in Fig. 10, a misjudgment occurred at the

beginning of the ponding which was 5 minutes ahead at Node 2. Three peaks appeared in the ponding process of Node 95,

and each one was identified. No ponding occurred at node 238 with the testing precipitation and the model predicted

correctly. Overall, the model prediction was relatively accurate at each node.







**Figure 10: Comparison between the predicted volume of ponding and simulation from the hydrodynamic model at the selected nodes.**

From the comparison of simulation from the hydrodynamic model and predicted results of the neural networks as shown in the charts and tables above, it was reliable to judge the occurrence and predict the location and process of ponding using the LSTM-based model trained by the simulation from the hydrodynamic model.

## 3.2 Model correction

The model in this paper was trained based on the simulation results from the hydrodynamic model. However, the difference
between the simulation from the hydrodynamic model at the monitoring points and the actual measured monitoring data persisted during the actual operation of the pipe network, which inevitably degraded the accuracy of the LSTM-based model on ponding forecasting. To compensate for this difference and improve the accuracy, the model was corrected by the measured rainfall data, level or flow data at the monitoring points, and the ponding data.





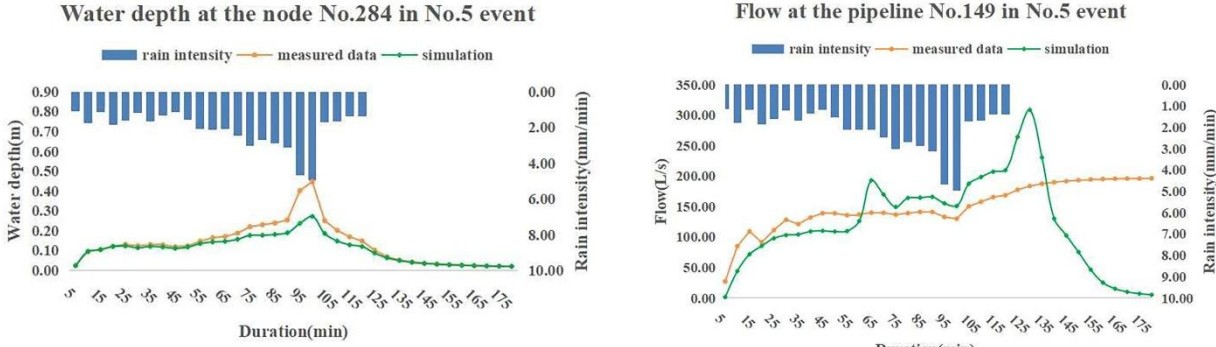

**Figure 11: Comparison between the measured data and simulation from the hydrodynamic model at Node 284 and Pipeline 149 in the case area.**

As shown in Fig. 11, rainfall event No.5 was one of the measured precipitation in the case area used to evaluate the performance of the corrected model where the maximum precipitation intensity reached 4.97mm/min. In this event, the measured flow or level data at the monitoring points and the simulation from the hydrodynamic model using the measured precipitation were compared to present the discrepancy between the measured and simulated results. Fig. 11 showed the discrepancy at the monitoring node 284 and the monitoring pipeline 149 existing, for example.

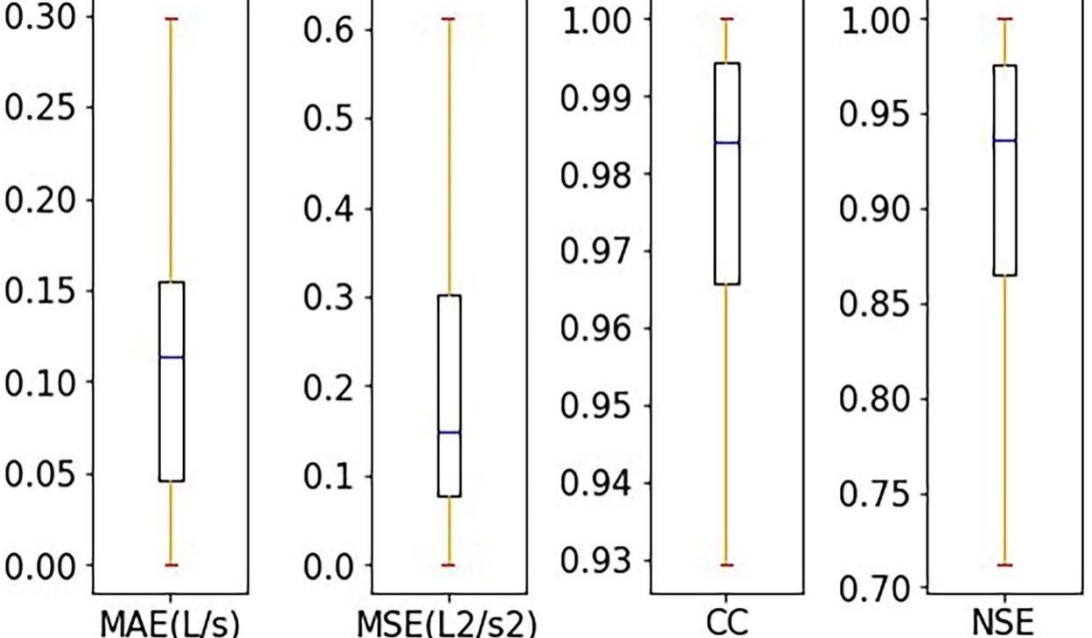

**Figure 12: Box plots of mean score values on the test set for all nodes in the model updating procedure.**



The ponding process predicted by the corrected model and the actual monitoring ponding data were compared to evaluate the model performance. The approach to correcting the model was reliable if the two were in good agreement. Fig. 12 illustrated the overall performance of the corrected model of all nodes on the ponding volume forecasting.

Fig. 12 showed the range of the mean score values on the test set of all nodes after being updated. As shown by the figure, the median values of CC and NSE scores maintained above 0.98 and 0.9 respectively, while the maximum value of MAE scores remained lower than 0.30L/s and that of MSE was lower than $0.6L^2/s^2$.

As shown in Table 7, the NSE score at each node was stably above 0.9, and the CC score was above 0.95, which suggested that the corrected model performed well on the test set, and the performance of the corrected model at each node was reliable. To justify the necessity for model updating and illustrate the performance of the model before and after correction more intuitively, Table 8 summarized the mean scores of all nodes for 5 measured rainfall events on the model performance before and after correction.

**Table 7: Mean score values on the test set of the 6 selected nodes on predicting the volume of ponding in the model updating procedure.**

| Node No. | MAE(L/s) | MSE($L^2/s^2$) | CC | NSE |
|---|---|---|---|---|
| 2 | 0.0912 | 0.1604 | 0.9774 | 0.9545 |
| 24 | 0.1359 | 0.2426 | 0.9863 | 0.9586 |
| 95 | 0.2916 | 0.7378 | 0.9583 | 0.9071 |
| 238 | 0.0855 | 0.1842 | 0.9773 | 0.9302 |
| 313 | 0.0642 | 0.0571 | 0.9896 | 0.9727 |
| 335 | 0.0943 | 0.1135 | 0.9942 | 0.9683 |

All the score values of the four indicators from the corrected model were much better than those from the uncorrected model in Table 8, especially specifically the NSE score before the correction was less than 0, as compared to 0.8316 after correction. It proved the necessity for model updating.

**Table 8: Mean score values of all nodes for 5 measured events from the corrected model versus those from the uncorrected model.**

| | MAE(L/s) | MSE($L^2/s^2$) | CC | NSE |
|---|---|---|---|---|
| Before updating | 0.5719 | 4.5045 | 0.1139 | < 0 |
| After updating | 0.1504 | 0.5919 | 0.9309 | 0.8316 |

Fig. 13 plotted the specific ponding process at the selected nodes in the measured rainfall event No.5 predicted by the corrected model as compared to those predicted by the model before correction.

Fig. 13 showed that the revised model performed better at all the selected nodes in terms of not only starting and ending time but also the process of ponding than the uncorrected model. The prediction by the corrected model had a high consistency with the measured ponding volume at each node in Fig. 13, which proved the reliability of the approach to introducing the

measured monitoring data to correct this LSTM-based model as well.



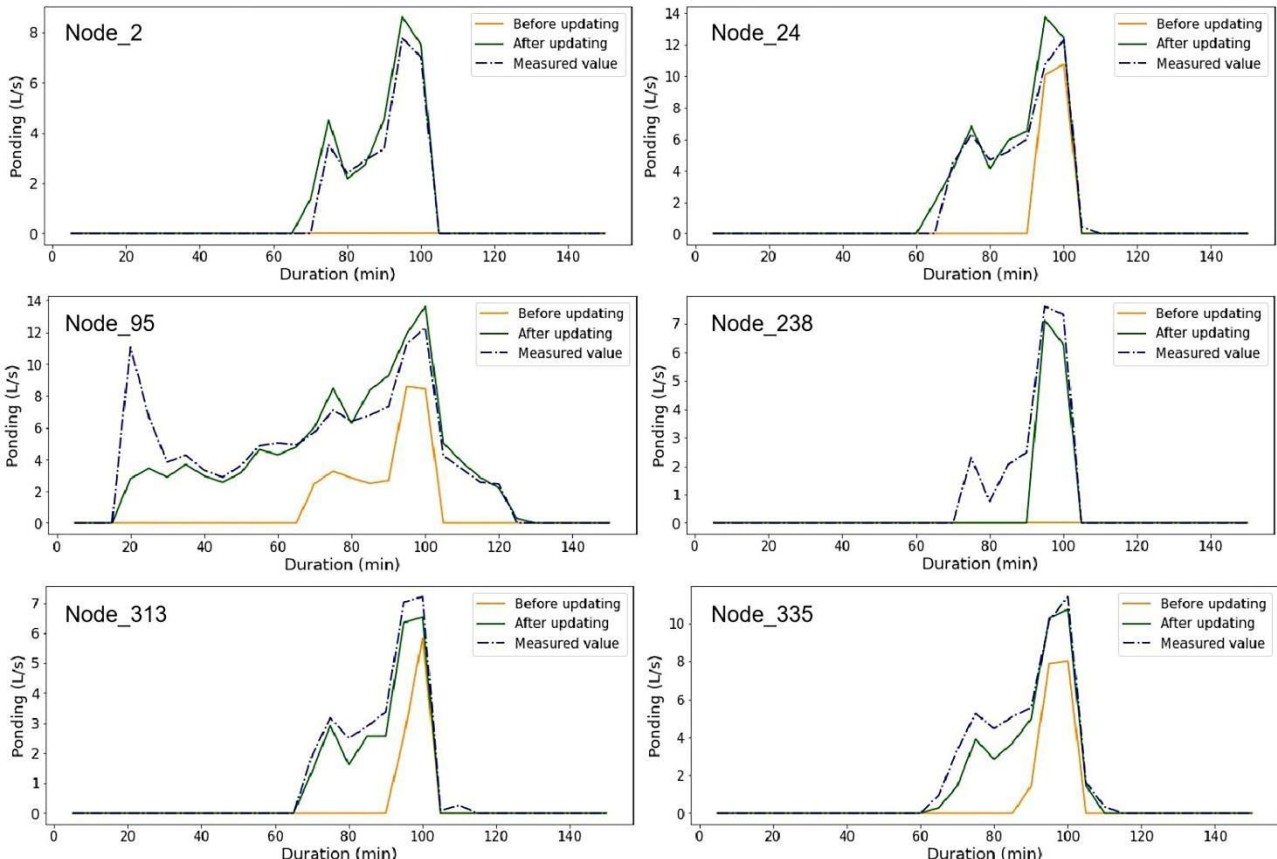

**Figure 13: Comparison between the predicted volume of ponding and measured values at the selected nodes in rainfall event No.5.**

By the comparison between the prediction by the model before and after correction, it could be concluded that it was
necessary to correct the model according to the measured monitoring data. The performance of the revised model was
relatively reliable on the ponding volume forecasting at any node.



# 4 Discussion

## 4.1 Comparison of neural network structures

Figure 14: Schematic diagrams of different network structures for comparison.

To verify the superiority of the variant of LSTM structure in the flow confluence process proposed in this study (model A), it was compared with the conventional LSTM structure (model B) in Fig. 14(a ~ c). Model B had the same structure as model A, which was divided into two processes, the runoff process and flow confluence process, and the runoff process in model B was the same as that in model A, which was presented in Fig. 14(a). However, the details of network architecture in the flow



confluence process of model B were different from that of model A (b ~ c), where there was no multi-task learning mechanism in its learning process.

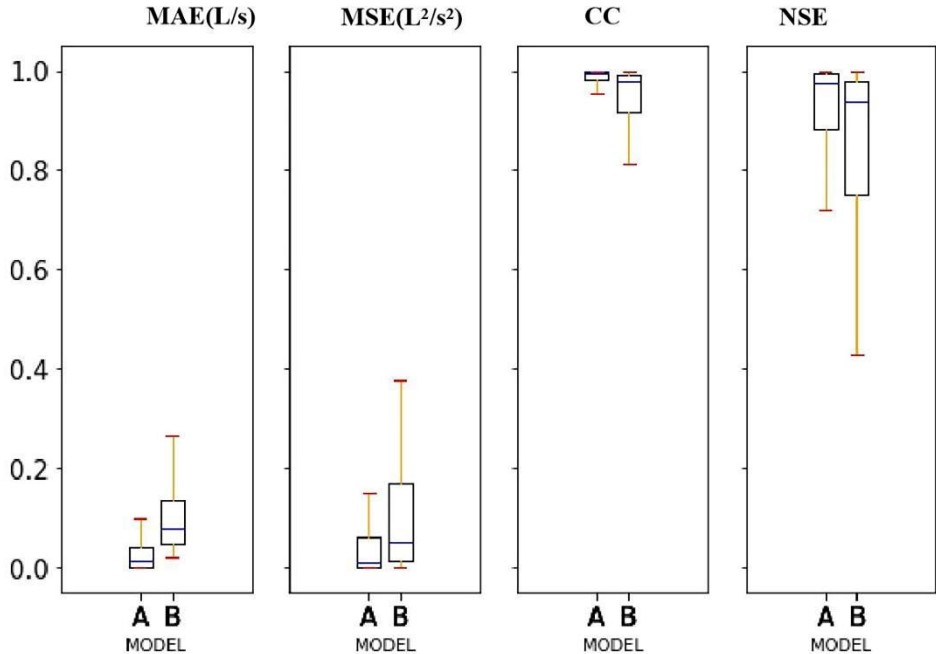

**(a) The results of the proposed model A as compared to those obtained from model B.**

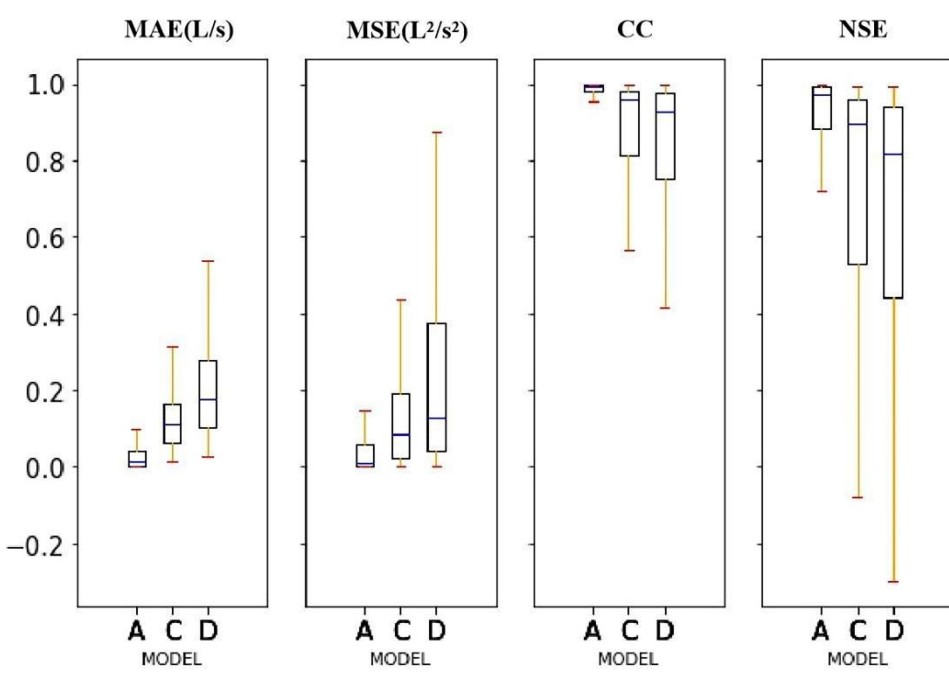




**(b) The results of the proposed model A as compared to those obtained from models C and D.**

**Figure 15: Comparison of model performance on the ponding volume forecasting.**

Fig. 15(a) presented the range of mean score values on the test set of all nodes on the performance of the model A and B on the ponding volume forecasting. As shown in Fig. 15(a), the range of the MAE and MSE scores of model A was respectively

half of that of model B. CC scores of model A varied in a small range and were close to 1, while those of model B varied from about 0.8 to 1. NSE scores of model A were higher than 0.7 while those of model B were higher than about 0.4. It was obvious that the structure in Fig. 14(b) was superior to that in Fig. 14(c) when it was applied to forecasting ponding volume. Furthermore, to illustrate the necessity of two processes in tandem, the runoff process and flow confluence process, model A proposed in this paper was compared with models which obtained ponding information directly from rainfall data without

extracting the characteristics of the lateral inflows, i.e., LSTM (model C) and CNN (model D) as shown in Fig. 14(d) and Fig. 14(e), respectively.

Fig. 15(b) presented the range of mean score values on the test set of all nodes on the performance of the model A, C, and D on the ponding volume forecasting. As shown in Fig. 15(b), the range of the MAE and MSE scores of models A, C, and D almost expanded gradually while the trend in the variation of the CC and NSE scores was the opposite. It was shown that the

structure of two processes in tandem in model A had an advantage over the structure in model C and the LSTM structure was better than CNN in dealing with these time series data.

**Table 9: Mean score values of all nodes obtained from models for predicting the volume of ponding.**

| Model | MAE(L/s) | MSE(L²/s²) | CC | NSE |
|:---:|:---:|:---:|:---:|:---:|
| A | 0.0309 | 0.1624 | 0.9960 | 0.9462 |
| B | 0.0622 | 0.1815 | 0.9578 | 0.8552 |
| C | 0.0849 | 0.2584 | 0.8823 | 0.7424 |
| D | 0.1358 | 0.3480 | 0.9257 | 0.7391 |

In a summary, the score values of model A varied in a smaller range and were closer to the optimal values than those of the other 3 alternatives in Fig. 15, which suggested a better performance of the proposed LSTM-based model (model A). This

was also supported by Table 9, which showed the mean score values of all nodes of the four models on the test set. According to the table, the performance ranking of the four models was model A > model B > model C > model D.



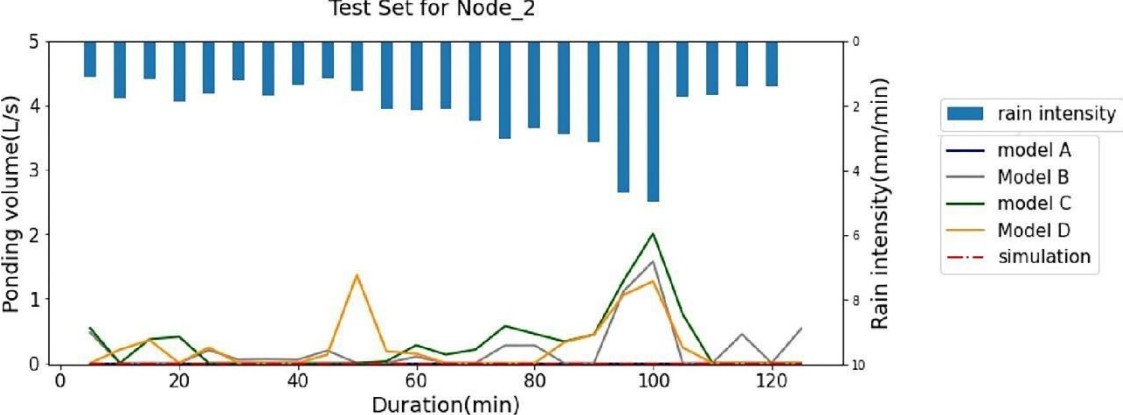

**(a) Case a where ponding did not occur at Node 2.**

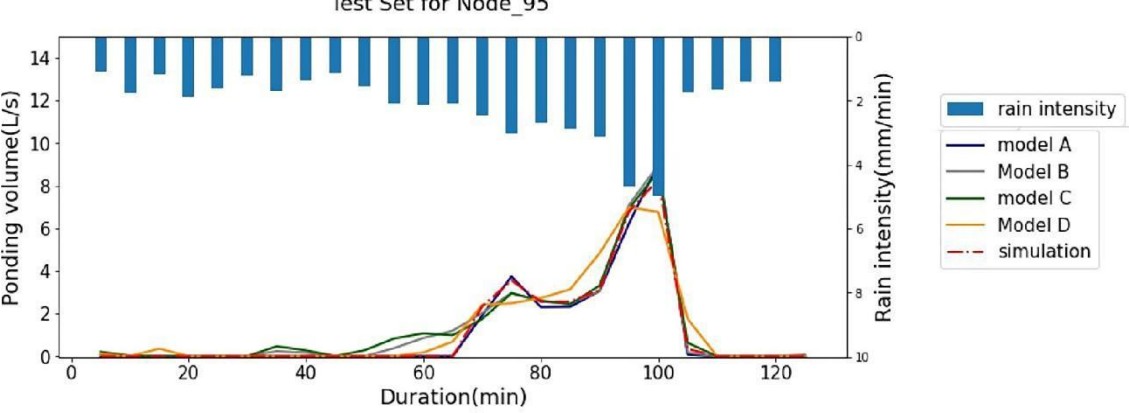

**(b) Case b where ponding occurred at Node 95.**

**Figure 16: Comparison between the predicted volume of ponding by the LSTM-based model (model A) and models B, C, and D in a certain rainfall event.**

Fig. 16 showed two cases where ponding did not occur at Node 2 (a) and occurred at Node 95 (b) respectively in a testing rainfall event. In case a, about 2-4L/s ponding volume was falsely reported by all three alternative models when ponding did

not occur. It pointed out that the reliability of these models was far worse than that of model A, where the introduced classification layer helped to judge whether the node was flooded or not. In case b where ponding lasted for 40 minutes, model A could describe the ponding process more accurately compared to the other alternatives.

The comparative analysis indicated that the LSTM-based model proposed in this paper had remarkable superiority over the other 3 alternatives in ponding volume forecasting because of the following two reasons: one of them was the two processes

in tandem, the runoff process, and the flow confluence process, and the other one was the variant of LSTM structure in the flow confluence process. The first one reduced the computational burden of this data-driven model and avoided interference



with each other when training separately; the second introduced an auxiliary task to judge whether ponding occurred and tried to get rid of the interference of the time points without ponding on the ponding volume forecasting.

**4.2 The influence of the number of monitoring points on the model modification**

In the model correction procedure, it was easy to spot from the trial that the effect of model modification depended on whether the current layout of the monitoring points reflected the hydraulic conditions of the pipe network. An unreasonable layout of the monitoring equipment might lead to a failure of model modification.

There were in total 15 level gauges and 3 flowmeters in the current pipe network; the layout of the monitoring points was shown in Fig. 7(b). Different numbers of the monitoring points were randomly selected from the current monitoring sites as

a quantitative control group to analyze whether and how the number of monitoring points impacted the performance of the revised model. Scores in Fig. 17 presented the evaluation results of the revised model on the ponding volume forecasting when different numbers of monitoring sites were selected to correct the model.

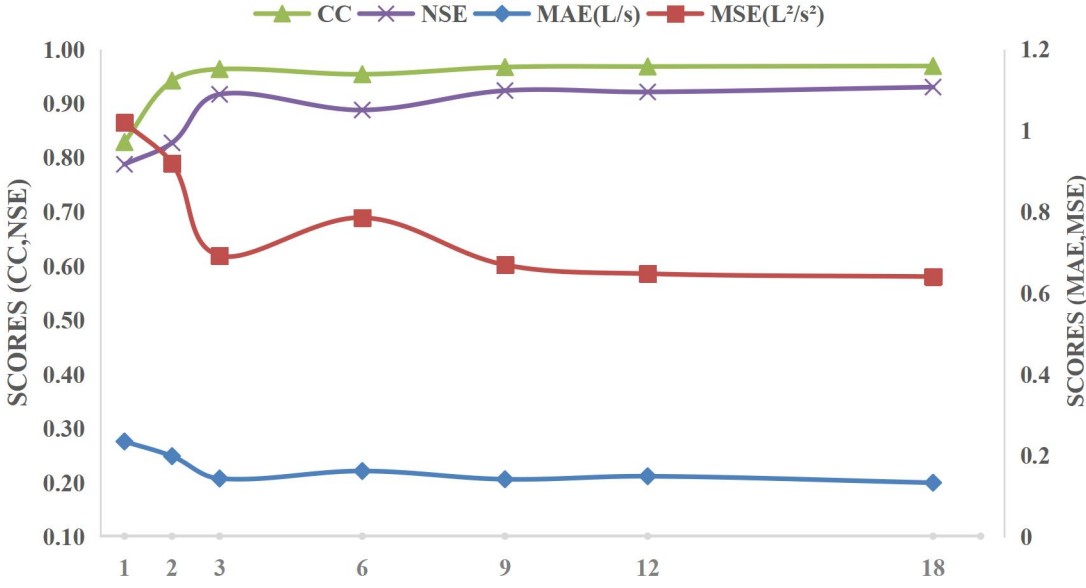

Figure 17: Score values of different numbers of monitoring points on the performance of the corrected model.

NSE scores stayed around 0.9 when the number of the monitoring points exceeded 6 and other scores were almost consistent with NSE. It turned out that increasing the number of monitoring points had little influence on the accuracy of the corrected model when the number of the monitoring sites was over one per hectare. However, when the number of





monitoring points went down to about 0.5 per hectare, i.e., the number of monitoring points was less than 3, the results of the revised model became incredibly bad, for example, the NSE score was lower than 0.8 when the number of monitoring points

was only 1.

It could be concluded that if the number of monitoring sites was less than one per hectare, the performance of the revised model could not be guaranteed, otherwise, if the number of monitoring sites was one per hectare or even more, the accuracy of the revised model on the ponding volume forecasting could be much higher.

**5 Conclusions**

An optimized LSTM-based model applied to early warning and forecasting of ponding in the urban drainage system was proposed in this paper. According to the research results, the main conclusions of this study are summarized as follows:

1 The LSTM-based model is constructed based on a relatively accurate hydrodynamic model. However, the difference between the simulation from the hydrodynamic model and the measured monitoring data persists, which leads to a discrepancy between the predicted results from the proposed LSTM-based model and the actual situation. Therefore, by

425 calibration using real-life data, the LSTM-based model can closely reflect the actual situation of the pipe network. The revised model realizes the real-time prediction of ponding with a certain accuracy in the case pipe network and the predicted information includes the location of ponding and the process of ponding at each node.

2 The model is split into two tandem processes, the first is to simulate the process of lateral inflows entering the nodes and the second is to predict the volume of ponding at each node in the pipe network. And a multi-task learning mechanism is

introduced to optimize the network structure in the flow confluence process. These two structures greatly improve the performance of the LSTM-based model. The advantage of the proposed variant of LSTM (model A) in handling the task of ponding volume forecasting is demonstrated by a comparison with the conventional LSTM structure (model B). Then, as compared to the models based on LSTM and CNN structures respectively (model C and D), the construction with two processes in tandem based on LSTM is demonstrated to be reliable.

3 The performance of the corrected model on ponding volume forecasting is reliable if the number of the monitoring sites is over one per hectare. Increasing the number of monitoring points further has little impact on the performance of the model.

Overall, the LSTM-based approach provides a possibility for early warning and forecasting of ponding in the urban drainage system and has a remarkable improvement after being corrected by the measured monitoring data. In this paper, all operations on model correction were carried out in an off-line mode. Real-time online correction should be further explored

in future studies. Besides, the model will be further optimized by considering the variation of ponding areas with the two-dimensional groundwater overflow process integrated into the model.

*Code availability.* The pieces of code that were used for all analyses are available from the authors upon request.




*Data availability.* All data used in this study are available from the authors upon request.

*Competing interests.* The contact author has declared that neither they nor their co-authors have any competing interests.

*Acknowledgment.* The authors would like to appreciate all of the team members for their insightful comments and
constructive suggestions to polish this paper in high quality.

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
