# Peer review of "An optimized LSTM-based approach applied to early warning and forecasting of ponding in the urban drainage system"

_EGUsphere, 2022_

## Author Comment (AC2)

Dear Reviewers,

Thank you very much for your time involved in reviewing the manuscript and your very encouraging comments on the merits.

**Comments:**

*"The authors proposed an LSTM-based emulator to simulate the ponding process in the drainage system, which is critical to urban flooding study. The emulator is composed of two LSTM models to sequentially simulate node lateral flows and the ponding volume, followed by a correction model. The proposed emulator was successfully applied to a case study and showed superior performances over some simplified versions (e.g., a lumped model using LSMT/CNN). I appreciate the hard work that has been put in by the authors. However, I have the following concerns which might require further revision before the manuscript can be accepted."*

We also appreciate your clear and detailed feedback and hope that the explanation has fully addressed all of your concerns. In the remainder of this letter, we discuss each of your comments individually along with our corresponding responses.

To facilitate this discussion, we first retype your comments in italic font and then present our responses to the comments.

**Comment 1:**

*First of all, I had a hard time following the manuscript. Readability is critical to a renowned journal such as HESS. The current status of the manuscript does not meet the requirement. For example, there are a lot of run-on sentences. A rule of thumb is that the length of a sentence does not exceed two lines. Coherence is also an issue. Many sentences are 'loosely' connected in a logical sense. It would be a pity if the message is not clearly communicated while so much work has been done. I suggest the authors further greatly revise the language (a professional English editor might help in this case).*

**Response 1:**

Thanks for your suggestion on improving the accessibility of our manuscript. We have revised the language by a professional English speaker. We have split the run-off sentences, and revised the logically incoherent sentences.

**Comment 2:**

*The second issue is associated with the model CR of the LSTM-based emulator (btw, what is CR abbreviated for?). I don't quite understand the descriptions of the model CR (i.e., L153-166). Neither Figure 6 is illustrative to me. Do the monitoring data refer to the measured lateral flows at the monitored nodes? Is the correction model trained on pairs of simulations and monitored measurements or based on a pre-trained mapping (i.e., using transfer learning)? Please specify.*

**Response 2:**

Thanks for your comment. CR is abbreviated for correction of the runoff process. Monitoring data refer to the measured water depths at the monitored nodes and flows at the monitored pipelines.

Model CR is designed to update the runoff process in the primary LSTM-based model. The correction has two steps: training and updating. Firstly, the model CR is trained based on a pre-trained mapping from X to Y (as shown in Fig.6). Then, it is updated on pairs of measured rain data, monitored water depths and flows.

We have adjusted Figure 6 in our paper. The related contents are also provided below for your quick reference. See L135, L146-147 and L150-152 for details.

[Figure]

**Figure 6:The architecture of Model CR. (MLP -- Multi-Layer Perception)**

**Comment 3:**

*The last concern is the mass balance of the emulator. Though not an expert in urban drainage systems, I consider that the mass conversation plays a key role in balancing the water exchanges between nodes. Does the proposed LSTM model account for that? If not, please specify the reason for not doing this.*

**Response 3:**

Thanks for your comment. The proposed LSTM model does account for the mass conversation between nodes. The model is trained using data simulated by mobilizing the SWMM model. The hydrodynamic model mainly simulates the flow of runoff and external flows in the pipes, channels and etc, including overflow, outflow and transmission of the pipe network system. The water exchanges between nodes have been covered in the simulation process.

**Comment 4:**

*Other minor edits:*

*L37-40: The authors point out the importance of the dataset. I'm wondering whether the author performed a sort of convergence test to evaluate how much data is sufficient for the proposed LSTM emulator training.*

**Response 4:**

Thanks for your comment. The related supplement has been added in our paper and are also provided below for your quick reference. See L210-216 for details.

"In general, a small training set normally leads to poor approximation effect. Thus a convergence test

was performed to evaluate how much data was required for the proposed LSTM-based model to obtain the desired approximation effect. The model performances using different sizes of training data were compared, as shown in Fig. 9. When the data size was reduced to 2/3 of the origin volume, the model performance fell down to 90% of the original. And if the data size was halved, less than 80% of the origin model performance remained. "

[Figure]

**Figure 9: The learning curve which describes the relationship between model performance and data volume.**

**Comment 5:**

*L50: 'not discussed' --> 'not explored'; 'not available' --> 'not feasible'*

*L77: 'influencing' --> 'influential'*

*L195: 'tb and ta is' --> 'ta and tb are'*

*L226: I like the usage of hyperopt here.*

*L338: 'In a summary' --> 'In summary'*

**Response 5:**

Thanks for your suggestion on improving the accessibility of our manuscript. These questions have been revised one by one.

**Comment 6:**

*L82: 'MAE, MSE, CC, NSE' --> We usually put full names before abbreviations*

**Response 6:**

Thank you for your comment. The related content have been added in our paper. See L85-87 for details. The related contents are also provided below for your quick reference.

"(MAE -- Mean Absolute Error, MSE -- Mean Squared Error, CC -- Correlation coefficient, NSE --

Nash-Sutcliffe efficiency coefficient)"

**Comment 7:**

*Figure 2 caption: '... test process in the runoff process' --> '... test procedures in developing the LSTM-based runoff emulator'. Also, many captions are too brief to provide enough information about these complicated figures.*

**Response 7:**

Thank you for your comment. The related content have been revised in our paper. See L96, L114, L135, L206, L306-307, L354-356, L363-364 for details. They are also provided below for your quick reference.

Figure 2: Figure 2: The training, validation, and testing procedures used when developing the LSTM-based runoff emulator. (MAE -- Mean Absolute Error, MSE -- Mean Squared Error, CC -- Correlation coefficient, NSE -- Nash-Sutcliffe efficiency coefficient)

Other captions have also been modified as follows:

Figure 3: The network structure of the flow confluence process (for a single node).

Figure 4: The error transmission during training from the runoff process to the flow confluence process.

Figure 5: Model correction system. CR is abbreviated for correction of the runoff process.

Figure 8: A demonstrative example to show the effect of adding white noise.

Figure 12: Comparison between the predicted ponding volume and simulation from the hydrodynamic model at the selected nodes in several testing rainfall events randomly chosen.

Figure 16: Schematic diagrams of different network structures for comparison. (a) The same runoff process in model A and B. (b) The multi-target learning in the flow confluence process of model A marked light blue. (c) The flow confluence process in model B marked dark blue. (d) The LSTM in model C. (e) The CNN in model D.

Figure 17: Comparison of model performance on the ponding volume forecasting. The results of the proposed model A are compared to those obtained from models B, C, and D.

**Comment 8:**

*Figure 3: For each of the two emulated processes, is only one LSTM used for all nodes? Or, is a separate LSTM used for each node?*

**Response 8:**

Thanks for your comment. For each of the two emulated processes, a separate LSTM is used for each node.

**Comment 9:**

*L91-93: That's a super long sentence and there are a lot!*

**Response 9:**

Thank you for your comment. The long sentence has been revised and is also provided below for your quick reference. See L89-91 for details.

"The flow confluence process is set up in the same manner as the simulation process of a hydrodynamic model (e.g., the SWMM model). If we compare the urban drainage system to a black box, only the lateral inflows at each node and outflows from the outlets enter and leave the system, respectively (Archetti et al., 2011)."

**Comment 10:**

*L100-102: Are the classification module and OUT_MODULE also two MLPs?*

**Response 10:**

Thanks for your comment. The 'CLASSIFICATION MODULE' and 'OUT MODULE' are two separate MLPs as the outputs for two tasks respectively.

**Comment 11:**

*L105-106: I don't understand which layer in the LSTM module is shared by the classification and out modules.*

**Response 11:**

Thanks for your comment. We have revised our paper. The related contents are also provided below for you quick reference. See L105-108 for details.

"Moreover, the multi-task learning has a hard parameter sharing mechanism, which effectively alleviate the over-fitting of the model. The parameters in the 'LSTM_MODULE' (including the parameters of the LSTM layers, batch normalization layers, activation functions, etc.) are shared by the 'CLASSIFICATION_MODULE' and ´OUT_MODULE'."

**Comment 12:**

*L116-119: To evaluate the impact of the gaussian filter, is there a comparison between the current emulator and one without the gaussian noising procedure?*

**Response 12:**

Thanks for your comment. We have revised the related contents and provided them below for your quick reference. See L97-100 for details.

"As illustrated in the pink block in Figure 3, a Gaussian layer is added after the input layer in the flow confluence process during training. The gaussian layer serves as a filter to compensate for the inaccuracy of the prediction (by the hydrodynamic model) in the runoff process. The model is trained to minimize the differences between the predictions (from the neural network, i.e. the output from the runoff process) and the simulations (from the hydrodynamic model)."

**Comment 13:**

*Eqs(1)-(4): I suggest moving the calculation of the error term to the appendix to improve the readability.*

**Response 13:**

Thank you for your comment. However, we believe that this part is very important in our model. The error is to explain the performance of the runoff process, and to compensate for the difference between the input of the flow confluence process in the actual application and the simulation data used in the training. So we think that it is better to put the calculation of the error in the text.

**Comment 14:**

*Eq.(5): What is 'lg'? Please use 'log' if you mean logarithm operation.*

**Response 14:**

Thank you for your comment. We have revised Eq.(5) in our paper. See L182 for details. It is also provided below for your quick reference.

$$q = \frac{167A(1+C\log P)}{(t+b)^n} = \frac{1600(1+0.846\log P)}{(t+7.0)^{0.656}} \tag{5}$$

**Comment 15:**

*L197: Is Pilgrim & Cordery a reference? If yes, please provide the year.*

**Response 15:**

Thanks for your comment. The related reference has been added to our paper. The related contents are also provided below for your quick reference. See L190-191 for details.

"The Pilgrim & Cordery was a method to count the historical rainfall data, and deduce the rainstorm pattern from it (Pilgrim and Cordery, 1975)."

**Comment 16:**

*L229-231: missing subjects of the two sentences.*

**Response 16:**

Thank you for your comment. We have revised these two sentences in our paper. See L237-238 for details. They are also provided below for your quick reference.

"Then, it received a random combination of hyper-parameters from the search space to do the training and returned the training loss. It used the built-in search algorithms like the tree of Parzen estimators (TPE) to determine the following hyper-parameters combination."

**Comment 17:**

*Table 3: What are the optimal hyperparameters of the MLP used for model CR? i.e., the number of neurons in each layer and the number of hidden layers. How about the hyperparameters of the classification and out modules?*

**Response 17:**

Thanks for your comment. We have revised Table 3 in our paper. The related contents are also provided below for your quick reference. See L234-236 for details.

**Table 3: Hyper-parameters configuration in model setup and correction processes.**

|  | Hyper-parameters | Model CR | | Flow confluence process |
|---|---|---|---|---|
|  |  | Runoff process | Fine-tuning process |  |
| Model setup | Normalization | Z-score | Z-score | Min-Max |
|  | Batch size | 150 | 150 | 150 |
|  | Epoch | 300 | 300 | 300 |
|  | Learning rate | 1e-2 | 5e-3 | 1e-2 |
|  | Optimizer | Adam | SGD | SGD |
|  | LSTM hidden layer neurons | 16 | - | 256 |
|  | MLP hidden layer neurons | 16 | 1536/3072 | 256/128* |
|  | LSTM layers | 2 | - | 4 |
|  | MLP layers | 1 | 2 | 2* |
| Model correction | Learning rate | 1e-4 | | 5e-5 |
|  | Optimizer | Adam | | SGD |

Note: * means to set the hyperparameters of 'CLASSIFICATION_MODULE' and 'OUT_MODULE' in the flow confluence process to the same values.

"The column 'Fine-tuning process' in Table 3 lists the optimal hyperparameters of the MLP used for model CR. The number of MLP hidden layers is 2, and the numbers of neurons in each layer are 1536 and 3072 respectively. Besides, the hyperparameters of the classification and out modules in the flow

confluence process are set to the same values. The number of MLP hidden layers is 2 and the numbers of neurons in each layer are 256 and 128 respectively."

**Comment 18:**

*L274: Why are these six nodes selected? (also shown in Figure 9)*

**Response 18:**

Thanks for your comment. The related supplement has been added to the paper. The contents are also provided below for your quick reference. See L276-280 for details.

"The six nodes were selected according to their severity of ponding and the spatial distribution to show the difference in model performance between nodes. Three of them were chosen because their percentages of samples where ponding occurred in the training set were less than 50%, and others were opposite. For example, the percentage of Node 238 was 18.33%, while that of Node 95 was 98.6%. Moreover, Fig. 10 marks their locations."

**Comment 19:**

*Figure 10: It is the emulated ponding volume before the model correction or CR, right? If yes, why is it different from the lines labeled by 'Before updating' in Figure 11?*

**Response 19:**

Thanks for your comment. The ponding at the selected nodes in Figure 12 occurred in some designed rain events randomly selected from the test set. However, the lines labeled 'Before updating' in Fig.15 are drawn in the measured rainfall event No.5.

**Comment 20:**

*L336-341: Should these sentences be grouped into one paragraph?*

**Response 20:**

Thank you for your comment. We have grouped these sentences into one paragraph. See L341-345 for details. They are also provided below for your quick reference.

"Fig. 15 plotted the specific ponding process at the selected nodes in the measured rainfall event No.5 predicted by the corrected model compared to those anticipated before correction. Fig. 15 showed that the revised model performed better at all the selected nodes in terms of not only starting and ending time but also the process of ponding than the uncorrected model. The prediction by the corrected model had a high consistency with the measured ponding volume at each node in the figure, which proved the reliability of the approach to introducing the measured monitoring data to correct this LSTM-based model as well."

**Comment 21:**

*Figure 15: combining (a) and (b)?*

**Response 21:**

Thank you for your comment. We have revised Figure 17 in our paper. See L362-364 for details. It is also provided below for your quick reference.

[Figure]

Figure 17: Comparison of model performance on the ponding volume forecasting. The results of the proposed model A are compared to those obtained from models B, C, and D.

We would like to take this opportunity to thank you for all your time involved and for this great opportunity for us to improve the manuscript. We hope you will find this revised version satisfactory.

Sincerely,

The Authors

---

## Author Response (AR1)

Dear Editor and Reviewer,

On behalf of my co-authors, we thank you very much for giving us an opportunity to revise our manuscript, and we also appreciate reviewers very much for their positive and constructive comments and suggestions on our manuscript entitled "An optimized LSTM-based approach applied to early warning and forecasting of ponding in the urban drainage system".

We revised the manuscript according to these comments and suggestions. In general, we have tried our best to revise our manuscript and provide the point-by-point responses. All changes were marked using the "Track Changes" function in the revised manuscript. Attached please find our responses to the referees' comments.

Once again, thank you very much for your comments and suggestions. And we hope this revised manuscript has addressed your concerns, and look forward to hearing from you.

Thank you and best regards.

Sincerely,

The Authors

**Reply to Reviewer #1**

Dear Reviewers,

Thank you very much for your time involved in reviewing the manuscript and your very encouraging comments on the merits.

**Comments:**

*"This paper proposed an optimized LSTM-based model applied to early warning and forecasting of ponding in the urban drainage system. It can identify flooding locations and process of ponding quickly with relatively high accuracy. The research ideas and methods are well innovative."*

We also appreciate your clear and detailed feedback and hope that the explanation has fully addressed all of your concerns. In the remainder of this letter, we discuss each of your comments individually along with our corresponding responses.

To facilitate this discussion, we first retype your comments in italic font and then present our responses to the comments.

**Comment 1:**

*My main concern about this paper is related to the case area. The authors said "(Due to these structural characteristics) the performance of the model will not be limited by the size of the case area", but they only applied the proposed method to a small-scale case area (a residential district of 6.128 hm2). I think it would be necessary to explain the capability of the proposed method.*

**Response 1:**

Thanks for your suggestion on improving the accessibility of our manuscript. The explanation about the case area has been amended in the paper(See L159-163 of the article for details). The relevant contents are provided below for your quick reference.

"Due to this structural characteristic, the size of the case area does not limit the model's performance. Regarding the model structure, the output of the runoff process is the lateral inflow at a single node. Likewise, the output of the flow confluence process is the ponding volume at a single node. Regardless of the size of the pipe network, the output of the model is at each node. However, a large-scale pipe network with lots of nodes will significantly increase the time spent training the model and also require extra processing power."

**Comment 2:**

*Section 2.4.2 (Eq. 5) Why you used this formula to design rain intensity? This is the design formula used by the municipality (i.e. a routine in China), orï¼ Need specify.*

**Response 2:**

Thanks for your suggestion. The reason why we use Eq. (5) has been added to L176-179 in our paper. The relevant contents are also provided below for your quick reference.

"The rainstorm intensity for S city is designed using Eq. (5), which is obtained according to a universal design storm pattern proposed by Keifer&Chu. The storm pattern is broadly used both at home and abroad, and the generated storms are usually extreme enough to reflect the state of the pipe networks under the most unfavorable conditions (Skougaard Kaspersen et al., 2017)."

**Comment 3:**

*What is Pilgrim & Cordery? Any equations?*

**Response 3:**

Thanks for your comment. The supplement of the Pilgrim & Cordery method has been added to L190-191 in the paper. The related contents are also provided below for your quick reference.

"Pilgrim & Cordery is a method to count the historical rainfall data and deduce the rainstorm pattern from it (Pilgrim & Cordery, 1975)."

To determine the storm pattern, the duration is divided into several periods, for each rainfall event, the sequence number of each period is determined according to the rainfall in each period from large to small, where large rainfall corresponds to a small sequence number. Then, average the serial numbers of each period, calculate the percentage of each rainfall to the total rainfall in each period, and take the average percentage in each period.

**Comment 4:**

*Please show equations to explain how you added the noise as the description is not clear enough.*

**Response 4:**

Thanks for your comment. A supplement about the process of adding noise has been added to L199-206, and Figure 8 has also been added to the paper. The related contents are also provided below for your quick reference.

"Take the rainfall with a return period of 5a as an example. Fig. 8 shows the effect of adding noise, where the subfigure (1) shows the randomly generated Gaussian white noise over the duration, the subfigure (2) shows the distribution of reordered white noise, and the subfigure (3) zooms in on the part circled in (2). The subfigures (4) - (6) show the design rainfalls after adding 30%, 50%, and 70% white noise, respectively. Specifically, we have limited the noises near the rainfall peak, i.e., only negative noises are allowed there."

[Figure]

**Figure 8: A demonstrative example to show the effect of adding white noise.**

**Comment 5:**

*Why there are only 5 real-world rainfall events to verify the performance of the corrected model? If it is enough considering that you have 16960 synthetic rainfall events?*

**Response 5:**

Thanks for your comment. The relevant explanation is provided below for your quick reference.

During the model training, a great deal of synthetic rainfall events is intended to cover as many extreme weather conditions as possible. In this study, since we considered the simulation results of the verified hydraulic model as the ground truth, the model correction is only to fine-tune the trained weight parameters without requiring a large amount of measured data. Besides, in the process of model updating, the model is modified by reducing the learning rate on the measurement data set, where the size of the data set does not work.

**Comment 6:**

*It is recommended to add HESS's article to the reference*

**Response 6:**

Thanks for your suggestion. The supplementary references have been added to our paper. The relevant contents are also provided below for your quick reference.

**Supplementary References:**

Archetti, R., Bolognesi, A., Casadio, A. and Maglionico, M.: Development of flood probability charts for urban drainage network in coastal areas through a simplified joint assessment approach, Hydrology and Earth System Sciences, 15, 3115-3122, http://dx.doi.org/10.5194/hess-15-3115-2011, 2011.

100 Guo, K., Guan, M. and Yu, D.: Urban surface water flood modelling-a comprehensive review of current models and future challenges, Hydrology and Earth System Sciences, 25, 2843-2860, http://dx.doi.org/10.5194/hess-25-2843-2021, 2021.

Huong, H.T.L. and Pathirana, A.: Urbanization and climate change impacts on future urban flooding in Can Tho city, Vietnam, Hydrology and Earth System Sciences, 17, 379-394,
105 http://dx.doi.org/10.5194/hess-17-379-2013, 2013.

Moy De Vitry, M., Kramer, S., Dirk Wegner, J. and Leitao, J.P.: Scalable flood level trend monitoring with surveillance cameras using a deep convolutional neural network, Hydrology and Earth System Sciences, 23, 4621-4634, http://dx.doi.org/10.5194/hess-23-4621-2019, 2019.

Skougaard Kaspersen, P., Hoegh Ravn, N., Arnbjerg-Nielsen, K., Madsen, H. and Drews, M.:
110 Comparison of the impacts of urban development and climate change on exposing European cities to pluvial flooding, Hydrology and Earth System Sciences, 21, 4131-4147, http://dx.doi.org/10.5194/hess-21-4131-2017, 2017.

Yang, T., Hwang, G., Tsai, C. and Ho, J.: Using rainfall thresholds and ensemble precipitation forecasts to issue and improve urban inundation alerts, Hydrology and Earth System Sciences, 20,
115 4731-4745, http://dx.doi.org/10.5194/hess-20-4731-2016, 2016.

We would like to take this opportunity to thank you for all your time involved and for this great opportunity for us to improve the manuscript. We hope you will find this revised version satisfactory.

Sincerely,

120 The Authors

**Reply to Reviewer #2**

Dear Reviewers,

Thank you very much for your time involved in reviewing the manuscript and your very encouraging comments on the merits.

**Comments:**

*"The authors proposed an LSTM-based emulator to simulate the ponding process in the drainage system, which is critical to urban flooding study. The emulator is composed of two LSTM models to sequentially simulate node lateral flows and the ponding volume, followed by a correction model. The proposed emulator was successfully applied to a case study and showed superior performances over some simplified versions (e.g., a lumped model using LSMT/CNN). I appreciate the hard work that has been put in by the authors. However, I have the following concerns which might require further revision before the manuscript can be accepted."*

We also appreciate your clear and detailed feedback and hope that the explanation has fully addressed all of your concerns. In the remainder of this letter, we discuss each of your comments individually along with our corresponding responses.

To facilitate this discussion, we first retype your comments in italic font and then present our responses to the comments.

**Comment 1:**

*First of all, I had a hard time following the manuscript. Readability is critical to a renowned journal such as HESS. The current status of the manuscript does not meet the requirement. For example, there are a lot of run-on sentences. A rule of thumb is that the length of a sentence does not exceed two lines. Coherence is also an issue. Many sentences are 'loosely' connected in a logical sense. It would be a pity if the message is not clearly communicated while so much work has been done. I suggest the authors further greatly revise the language (a professional English editor might help in this case).*

**Response 1:**

Thanks for your suggestion on improving the accessibility of our manuscript. We have revised the language by a professional English speaker. We have split the run-off sentences, and revised the logically incoherent sentences.

**Comment 2:**

*The second issue is associated with the model CR of the LSTM-based emulator (btw, what is CR abbreviated for?). I don't quite understand the descriptions of the model CR (i.e., L153-166). Neither Figure 6 is illustrative to me. Do the monitoring data refer to the measured lateral flows at the*

 *monitored nodes? Is the correction model trained on pairs of simulations and monitored measurements or based on a pre-trained mapping (i.e., using transfer learning)? Please specify.*

**Response 2:**

Thanks for your comment. CR is abbreviated for correction of the runoff process. Monitoring data refer to the measured water depths at the monitored nodes and flows at the monitored pipelines.

Model CR is designed to update the runoff process in the primary LSTM-based model. The correction has two steps: training and updating. Firstly, the model CR is trained based on a pre-trained mapping from X to Y (as shown in Fig.6). Then, it is updated on pairs of measured rain data, monitored water depths and flows.

We have adjusted Figure 6 in our paper. The related contents are also provided below for your quick reference. See L135, L146-147 and L150-152 for details.

[Figure]

**Figure 6: The architecture of Model CR. (MLP -- Multi-Layer Perception)**

**Comment 3:**

*The last concern is the mass balance of the emulator. Though not an expert in urban drainage systems, I consider that the mass conversation plays a key role in balancing the water exchanges between nodes. Does the proposed LSTM model account for that? If not, please specify the reason for not doing this.*

**Response 3:**

Thanks for your comment. The proposed LSTM model does account for the mass conversation between nodes. The model is trained using data simulated by mobilizing the SWMM model. The hydrodynamic model mainly simulates the flow of runoff and external flows in the pipes, channels and etc, including overflow, outflow and transmission of the pipe network system. The water exchanges between nodes have been covered in the simulation process.

**Comment 4:**

*Other minor edits:*

*L37-40: The authors point out the importance of the dataset. I'm wondering whether the author performed a sort of convergence test to evaluate how much data is sufficient for the proposed LSTM emulator training.*

**Response 4:**

Thanks for your comment. The related supplement has been added in our paper and are also provided below for your quick reference. See L210-216 for details.

185 "In general, a small training set normally leads to poor approximation effect. Thus a convergence test was performed to evaluate how much data was required for the proposed LSTM-based model to obtain the desired approximation effect. The model performances using different sizes of training data were compared, as shown in Fig. 9. When the data size was reduced to 2/3 of the origin volume, the model performance fell down to 90% of the original. And if the data size was halved, less than 80% of the 190 origin model performance remained. "

[Figure]

**Figure 9: The learning curve which describes the relationship between model performance and data volume.**

**Comment 5:**

195 *L50: 'not discussed' --> 'not explored'; 'not available' --> 'not feasible'*

*L77: 'influencing' --> 'influential'*

*L195: 'tb and ta is' --> 'ta and tb are'*

*L226: I like the usage of hyperopt here.*

*L338: 'In a summary' --> 'In summary'*

200 **Response 5:**

Thanks for your suggestion on improving the accessibility of our manuscript. These questions have been revised one by one.

**Comment 6:**

*L82: 'MAE, MSE, CC, NSE' --> We usually put full names before abbreviations*

205 **Response 6:**

Thank you for your comment. The related content have been added in our paper. See L85-87 for details. The related contents are also provided below for your quick reference.

"(MAE -- Mean Absolute Error, MSE -- Mean Squared Error, CC -- Correlation coefficient, NSE -- Nash-Sutcliffe efficiency coefficient)"

210 **Comment 7:**

*Figure 2 caption: '... test process in the runoff process' --> '... test procedures in developing the LSTM-based runoff emulator'. Also, many captions are too brief to provide enough information about these complicated figures.*

**Response 7:**

215 Thank you for your comment. The related content have been revised in our paper. See L85-87, L96, L111, L135, L206, L306-307, L351-353, L375-376 for details. They are also provided below for your quick reference.

Figure 2: The training, validation, and testing procedures used when developing the LSTM-based runoff emulator. (MAE -- Mean Absolute Error, MSE -- Mean Squared Error, CC -- Correlation

220 coefficient, NSE -- Nash-Sutcliffe efficiency coefficient)

Other captions have also been modified as follows:

Figure 3: The network structure of the flow confluence process (for a single node).

Figure 4: The error transmission during training from the runoff process to the flow confluence process.

Figure 5: Model correction system. CR is abbreviated for correction of the runoff process.

225 Figure 8: A demonstrative example to show the effect of adding white noise.

Figure 12: Comparison between the predicted ponding volume and simulation from the hydrodynamic model at the selected nodes in 6 testing rainfall events that were chosen randomly.

Figure 16: Schematic diagrams of different network structures for comparison. (a) The same runoff process in model A and B. (b) The multi-target learning in the flow confluence process of model A,

230 marked in light blue. (c) The flow confluence process in model B marked in dark blue. (d) The LSTM structure in model C. (e) The CNN structure in model D.

Figure 18: Comparison of model performance on the ponding volume forecasting. The results of the

proposed model A are compared to those obtained from models B, C, and D.

**Comment 8:**

235   *Figure 3: For each of the two emulated processes, is only one LSTM used for all nodes? Or, is a separate LSTM used for each node?*

**Response 8:**

Thanks for your comment. For each of the two emulated processes, a separate LSTM is used for each node.

240   **Comment 9:**

*L91-93: That's a super long sentence and there are a lot!*

**Response 9:**

Thank you for your comment. The long sentence has been revised and is also provided below for your quick reference. See L89-91 for details.

245   "The flow confluence process is set up in the same manner as the simulation process of a hydrodynamic model (e.g., the SWMM model). If we compare the urban drainage system to a black box, only the lateral inflows at each node and outflows from the outlets enter and leave the system, respectively (Archetti et al., 2011)."

**Comment 10:**

250   *L100-102: Are the classification module and OUT_MODULE also two MLPs?*

**Response 10:**

Thanks for your comment. The 'CLASSIFICATION MODULE' and 'OUT MODULE' are two separate MLPs as the outputs for two tasks respectively.

**Comment 11:**

255   *L105-106: I don't understand which layer in the LSTM module is shared by the classification and out modules.*

**Response 11:**

Thanks for your comment. We have revised our paper. The related contents are also provided below for you quick reference. See L105-108 for details.

260   "Moreover, the multi-task learning has a hard parameter sharing mechanism, which effectively

alleviate the over-fitting of the model. The parameters in the 'LSTM_MODULE' (including the parameters of the LSTM layers, batch normalization layers, activation functions, etc.) are shared by the 'CLASSIFICATION_MODULE' and 'OUT_MODULE'."

**Comment 12:**

*L116-119: To evaluate the impact of the gaussian filter, is there a comparison between the current emulator and one without the gaussian noising procedure?*

**Response 12:**

Thanks for your comment. We have revised the related contents and provided them below for your quick reference. See L97-100 for details.

"As illustrated in the pink block in Figure 3, a Gaussian layer is added after the input layer in the flow confluence process during training. The gaussian layer serves as a filter to compensate for the inaccuracy of the prediction (by the hydrodynamic model) in the runoff process. The model is trained to minimize the differences between the predictions (from the neural network, i.e. the output from the runoff process) and the simulations (from the hydrodynamic model)."

**Comment 13:**

*Eqs(1)-(4): I suggest moving the calculation of the error term to the appendix to improve the readability.*

**Response 13:**

Thank you for your comment. However, we believe that this part is very important in our model. The error is to explain the performance of the runoff process, and to compensate for the difference between the input of the flow confluence process in the actual application and the simulation data used in the training. So we think that it is better to put the calculation of the error in the text.

**Comment 14:**

*Eq.(5): What is 'lg'? Please use 'log' if you mean logarithm operation.*

**Response 14:**

Thank you for your comment. We have revised Eq.(5) in our paper. See L180 for details. It is also provided below for your quick reference.

$$q = \frac{167A(1 + C\log P)}{(t+b)^n} = \frac{1600(1 + 0.846\log P)}{(t+7.0)^{0.656}}$$

(5)

**Comment 15:**

290 *L197: Is Pilgrim & Cordery a reference? If yes, please provide the year.*

**Response 15:**

Thanks for your comment. The related reference has been added to our paper. The related contents are also provided below for your quick reference. See L190-191 for details.

"The Pilgrim & Cordery was a method to count the historical rainfall data, and deduce the rainstorm
295 pattern from it (Pilgrim and Cordery, 1975)."

**Comment 16:**

*L229-231: missing subjects of the two sentences.*

**Response 16:**

Thank you for your comment. We have revised these two sentences in our paper. See L235-236 for
300 details.

"The hyper-parameters used in this paper were mainly determined by Hyperopt (BERGSTRA, J et al., 2013). Hyperopt is a Python library for hyper-parameter optimization that adjusts parameters using Bayesian optimization."

**Comment 17:**

305 *Table 3: What are the optimal hyperparameters of the MLP used for model CR? i.e., the number of neurons in each layer and the number of hidden layers. How about the hyperparameters of the classification and out modules?*

**Response 17:**

Thanks for your comment. We have revised Table 3 in our paper. The related contents are also
310 provided below for your quick reference. See L232-234 for details.

**Table 3: Hyper-parameters configuration in model setup and correction processes.**

| | Hyper-parameters | Model CR | | Flow confluence process |
|---|---|---|---|---|
| | | Runoff process | Fine-tuning process | |
| | Normalization | Z-score | Z-score | Min-Max |
| | Batch size | 150 | 150 | 150 |
| | Epoch | 300 | 300 | 300 |
| Model setup | Learning rate | 1e-2 | 5e-3 | 1e-2 |

| | Hyper-parameters | Model CR | | Flow confluence process |
|---|---|---|---|---|
| | | Runoff process | Fine-tuning process | |
| | Optimizer | Adam | SGD | SGD |
| | LSTM hidden layer neurons | 16 | - | 256 |
| | MLP hidden layer neurons | 16 | 1536/3072 | 256/128* |
| | LSTM layers | 2 | - | 4 |
| | MLP layers | 1 | 2 | 2* |
| Model correction | Learning rate | 1e-4 | | 5e-5 |
| | Optimizer | Adam | | SGD |

Note: * means to set the hyperparameters of 'CLASSIFICATION_MODULE' and 'OUT_MODULE' in the flow confluence process to the same values.

"The column 'Fine-tuning process' in Table 3 lists the optimal hyperparameters of the MLP used for model CR. The number of MLP hidden layers is 2, and the numbers of neurons in each layer are 1536 and 3072 respectively. Besides, the hyperparameters of the classification and out modules in the flow confluence process are set to the same values. The number of MLP hidden layers is 2 and the numbers of neurons in each layer are 256 and 128 respectively."

**Comment 18:**

*L274: Why are these six nodes selected? (also shown in Figure 9)*

**Response 18:**

Thanks for your comment. The related supplement has been added to the paper. The contents are also provided below for your quick reference. See L273-278 for details.

"The six nodes (as shown in Fig. 11) were selected because of the severity of consequence once ponding occurred, and also because they were relatively uniformly distributed in the pipe network. Moreover, three of them (Nodes 2, 238, and 313) were chosen because the positive samples (where ponding occurred) accounted for less than 50% of the training set, and the other three were in the opposite case. For example, at Node 238, the positive samples accounted for 18.33% of the training set, while at Node 95, up to 98.6% samples were positive."

**Comment 19:**

*Figure 10: It is the emulated ponding volume before the model correction or CR, right? If yes, why is it different from the lines labeled by 'Before updating' in Figure 11?*

**Response 19:**

Thanks for your comment. The ponding at the selected nodes in Figure 12 occurred in some designed

335 rain events randomly selected from the test set. However, the lines labeled 'Before updating' in Fig.15 are drawn in the measured rainfall event No.5.

**Comment 20:**

*L336-341: Should these sentences be grouped into one paragraph?*

**Response 20:**

340 Thank you for your comment. We have revised these sentences. See L339-342 for details. They are also provided below for your quick reference.

"To further demonstrate the effect of model correction procedure, we have shown the predicted ponding process at the 6 selected nodes for rainfall event No.5, obtained by using the model with and without correction, as shown in Fig. 15. As shown in the figure, the corrected model performed better

345 at all the selected nodes, e.g., more accurate prediction of start/end time of ponding, more accurate ponding curves (more similar to the measure ones)."

**Comment 21:**

*Figure 15: combining (a) and (b)?*

**Response 21:**

350 Thank you for your comment. We have revised Figure 18 in our paper. See L374-376 for details. It is also provided below for your quick reference.

[Figure]

Figure 18: Comparison of model performance on the ponding volume forecasting. The results of the proposed model A are compared to those obtained from models B, C, and D.

355

We would like to take this opportunity to thank you for all your time involved and for this great opportunity for us to improve the manuscript. We hope you will find this revised version satisfactory.

Sincerely,

The Authors

---

## Author Response (AR2)

Dear Editor and Reviewers,

On behalf of my co-authors, we thank you very much for giving us an opportunity to revise our manuscript, and we also appreciate reviewers very much for their positive and constructive comments and suggestions on our manuscript entitled "An optimized LSTM-based approach applied to early warning and forecasting of ponding in the urban drainage system".

We have tried our best to revise our manuscript according to these comments and suggestions and provide the point-by-point responses. All changes were marked using the "Track Changes" function in the revised manuscript. Attached please find our responses to the referees' comments.

Once again, thank you very much for your comments and suggestions. And we hope this revised manuscript has addressed your concerns, and look forward to hearing from you.

Thank you and best regards.

Sincerely,

The Authors

**Reply to Reviewer # 1**

Dear Reviewers,

Thank you very much for your time involved in reviewing the manuscript. We also appreciate your clear and detailed feedback and hope that the explanation has fully addressed all of your concerns. In the remainder of this letter, we discuss your comments along with our corresponding responses.

To facilitate this discussion, we first retype your comments in italic font and then present our responses to the comments.

**Comment 1:**

*"It is suggested that the conclusion of the paper needs to be further revised to balance the volume of the three conclusions, and highlight the main feature of the paper, how to apply deep learning method in the simulation of urban flooding."*

**Response 1:**

Thank you for comments. The conclusions have been revised in our paper. See L406-425 for details. They are also provided below for your quick reference.

"This work aims at promoting the application of deep learning in urban flood forecasting. Specifically, we have proposed an optimized LSTM-based approach in this study, which can quickly identify and locate ponding with relatively high accuracy.

According to the research results, the main conclusions of this study are summarized as follows:

The proposed model is constructed by two tandem processes (runoff process and flow confluence process) and utilizes a multi-task learning mechanism to achieve high accuracy. Over 15000 designed rainfall events were used for model training, which covers various extreme weather conditions. The median score of NSE for ponding forecasting is greater than 0.95, and the mean accuracy at any node to determine whether ponding occurs reaches higher than 0.98.

The superiority of the proposed model has been demonstrated by comparing with two widely used deep learning models: (traditional) LSTM and CNN models.

The superiority of the proposed model having two tandem processes is proved by comparing with LSTM and CNN structures with a single process. The mean NSE score for ponding volume forecasting of the proposed model is 0.9462, while that of LSTM and CNN structures with a single process is 0.7424 and 0.7391 respectively. Then, the superiority of the proposed model with a LSTM variant is demonstrated by a comparison with the conventional LSTM structure also with two tandem processes. As shown in Table 9, the mean NSE score of the latter is 0.8552.

An approach to model modification using real-life monitoring level and flow data is proposed in this paper. The proposed LSTM-based model is further calibrated to achieve better accuracy.

The LSTM-based model is corrected using two steps. First, the runoff process is corrected with the measured rain, level, and flow data referring to Parameter-based (Model-based) Transfer Learning. Then, the flow confluence process is updated using the updated lateral inflows at all nodes and the measured ponding volume. As shown in Table 8, the mean CC score at all nodes of the model with correction is 0.9309, while that of the model without correction is 0.1139."

We would like to take this opportunity to thank you for all your time involved and for this great opportunity for us to improve the manuscript. We hope you will find this revised version satisfactory.

Sincerely,

The Authors

**Reply to Reviewer #2**

Dear Reviewers,

Thank you very much for your time involved in reviewing the manuscript and your very encouraging comments on the merits.

**Comments:**

*"I would like to thank the authors for taking the effort in revising the manuscript to address my*
*comments. Though great revisions have been made, I have one more comment and some minor language revisions to suggest. I hope some of the edits would help the authors' future work."*

We also appreciate your clear and detailed feedback and hope that the explanation has fully addressed all of your concerns. In the remainder of this letter, we discuss each of your comments individually along with our corresponding responses.

To facilitate this discussion, we first retype your comments in italic font and then present our responses to the comments.

**Comment 1:**

*Section 2.3. -- Model correction: I still don't quite follow how the correction model has been developed, and there might be a miscommunication issue. Did you leverage an existing pre-trained model from*
*Pan et al (2010) to perform the correction here (L143)? If yes, how applicable is this pre-trained model to this study? Also, the sentence 'the model CR is trained based on a pre-trained mapping from X to Y' (L151) is confusing. It doesn't tell whether the CR model uses a trained model from another study (for the purpose of transfer learning) or is trained separately in this work. If it is trained separately, which I suppose was after the development of the two LSTM modules, why did you call it 'transfer learning'?*

**Response 1:**

Thanks for your comment. We have answered these questions one by one.

*Did you leverage an existing pre-trained model from Pan et al (2010) to perform the correction here (L143)? If yes, how applicable is this pre-trained model to this study?*

We did not leverage an existing pre-trained model from Pan et al (2010) to perform the correction here.
We have adjusted the reference to L141.

"Transfer learning is mainly used to transfer the knowledge of one domain (source domain) to another domain (target domain) such that the target domain can achieve better learning effects (PAN, S J, et al., 2010)."

*Also, the sentence 'the model CR is trained based on a pre-trained mapping from X to Y' (L151) is*

*confusing. It doesn't tell whether the CR model uses a trained model from another study (for the purpose of transfer learning) or is trained separately in this work.*

'A pre-trained mapping from X to Y' (L147) is the trained model in the runoff process introduced in Section 2.1.1.

*If it is trained separately, which I suppose was after the development of the two LSTM modules, why*
*did you call it 'transfer learning'?*

In the parameter-based(model-based) transfer learning, it is mainly assumed that some of the same parameters will be shared between the related tasks in the source and target domains, so that part of the network structure can be shared between the related tasks. In this paper, Y is the source domain, G is the target domain. The training process of the model CR between X and G introduces the network
structure and the trained parameters in the runoff process between X and Y, which is called 'transfer learning'.

**Comment 2:**

*Other minor edits in the introduction section:*
*L32: 'has-' --> 'has'*

*L33: I wouldn't call "deep learning as a form of training". Deep learning is a particular machine learning technique that leverages neural networks to learn nonlinear relationships from a dataset.*

*L38: 'some factors need to be improved' --> 'there are opportunities to further the application of deep learning ...'*

*L39: 'the dataset for training' --> 'the training dataset'*

*L39-40: 'There are studies utilizing deep learning algorithms for urban flood forecasting, but the developed model is trained on a small number of samples.' --> 'Many studies in urban flood forecasting only use a small number of samples to develop the deep learning models.'*

*L42-44: 'Secondly, monitoring equipment is expensive and thus not frequently available. Therefore, researchers have to rely on simulations produced from hydrodynamic models, however, often without*
*considering the accuracy of the models.' --> 'Secondly, due to the high cost of monitoring equipment, researchers usually have to rely on unvalidated simulations produced from hydrodynamic models.'*

*L47-48: "Such as ..." --> "Example includes but not limited to ..."*

*L54: "we propose an optimized LSTM-based approach, which is applied to early warning" --> "we propose an optimized LSTM-based approach for early warning ..."*

*L57: "(LSTM, CNN)." --> ", i.e., LSTM and CNN."*

*L58: "to achieve higher accuracy" --> "to improve the emulation performance"*

**Response 2:**

Thanks for your suggestion on improving the accessibility of our manuscript. These questions have been revised one by one.

**Comment 3:**

*L34: 'And unlike' --> 'Like'. Please avoid using 'and' in the beginning of the sentence, which is informal. (There are multiple cases throughout the manuscript. Please double check)*

**Response 3:**

Thank you for your comment. The related content have been revised in our paper. See L43, L135, L145,
L173, L204, L209, L252, L398 and L428 for details. They are also provided below for your quick reference.

L43: , and compared ... --> , then compared the ...

L135: , and how to ... --> . Moreover, how to ...

L145: 'And add multiple fully connected layers after ...' --> Then, multiple fully connected layers are
added after ...

L173: and the generated storms ... --> The generated storms ...

L204: and the ratios ... --> The ratios ...

L209: 'And if ...' --> Moreover, if ...

L252: 'And the subscript ...' --> The subscript ...

L398: 'And the other ...' --> Besides, the other ...

L428: 'And also, ...' --> Furthermore, ...

We would like to take this opportunity to thank you for all your time involved and for this great opportunity for us to improve the manuscript. We hope you will find this revised version satisfactory.

Sincerely,

The Authors